# TASAR: Transfer-based Attack on Skeletal Action Recognition

**Yunfeng Diao**[1]   **Baiqi Wu**[1,†]   **Ruixuan Zhang**[1]   **Ajian Liu**[2]   **Xiaoshuai Hao**[3]
**Xingxing Wei**[4]   **Meng Wang**[1]   **He Wang**[5,†]

[1]Hefei University of Technology   [2] Institute of Automation Chinese Academy of Sciences
[3] Beijing Academy of Artificial Intelligence   [4] Beihang University
[5] UCL Centre for Artificial Intelligence, Department of Computer Science, University College London

`diaoyunfeng@hfut.edu.cn, {2021214516, 2020217721}@mail.hfut.edu.cn,ajianliu92@gmail.com`

`xshao@baai.ac.cn, xxwei@buaa.edu.cn, eric.mengwang@gmail.com,he_wang@ucl.ac.uk`

## Abstract

Skeletal sequence data, as a widely employed representation of human actions, are crucial in Human Activity Recognition (HAR). Recently, adversarial attacks have been proposed in this area, which exposes potential security concerns, and more importantly provides a good tool for model robustness test. Within this research, transfer-based attack is an important tool as it mimics the real-world scenario where an attacker has no knowledge of the target model, but is under-explored in Skeleton-based HAR (S-HAR). Consequently, existing S-HAR attacks exhibit weak adversarial transferability and the reason remains largely unknown. In this paper, we investigate this phenomenon via the characterization of the loss function. We find that one prominent indicator of poor transferability is the low smoothness of the loss function. Led by this observation, we improve the transferability by properly smoothening the loss when computing the adversarial examples. This leads to the first **T**ransfer-based **A**ttack on **S**keletal **A**ction **R**ecognition, TASAR. TASAR explores the smoothened model posterior of pre-trained surrogates, which is achieved by a new post-train Dual Bayesian optimization strategy. Furthermore, unlike existing transfer-based methods which overlook the temporal coherence within sequences, TASAR incorporates motion dynamics into the Bayesian attack, effectively disrupting the spatial-temporal coherence of S-HARs. For exhaustive evaluation, we build the first large-scale robust S-HAR benchmark, comprising 7 S-HAR models, 10 attack methods, 3 S-HAR datasets and 2 defense models. Extensive results demonstrate the superiority of TASAR. Our benchmark enables easy comparisons for future studies, with the code available in the https://github.com/yunfengdiao/Skeleton-Robustness-Benchmark.

## 1 Introduction

S-HAR has been an important research topic in computer vision. Recently, S-HAR classifiers have been found to be susceptible to adversarial attack (Wang et al., 2021; Diao et al., 2021), suggesting adversarial attack potentially provides a useful tool for robustness tests for S-HAR classifiers. But not all attacks are equally practical. Existing S-HAR attacks are mainly proposed under white-box settings (Liu et al., 2020a; Tanaka et al., 2022), where the attacker has full access to the victim model's architecture, weights, and training details, or under query-based black-box settings (Diao et al., 2021; Kang et al., 2023b), where the attacker can make numerous queries (Diao et al., 2024a). However, neither approaches are impractical in real-world scenarios (*e.g.* autonomous driving (Guo et al., 2024), intelligent surveillance (Garcia-Cobo & SanMiguel, 2023) and human-computer interactions (Wang et al., 2020)), where either accessing the victim model or numerous queries is not attainable. Therefore, transfer-based attack, *i.e.* generating adversarial examples by attacking a surrogate model and then transfer them to target black-box models, is proposed as a promising alternative (Dong et al., 2018; Wang et al., 2021).

---

[†] Corresponding author

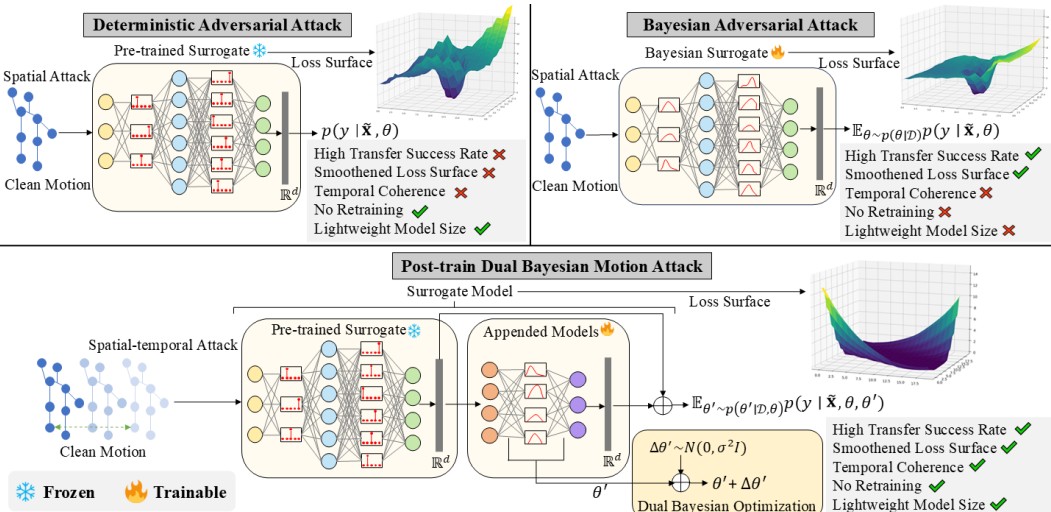

Figure 1: A high-level illustration of our proposed method. Results marked with a 'check mark' ($\sqrt{}$) indicate superior performance compared to those marked with a 'cross' ($\times$). Spatial attack: treats each frame independently. Spatial-temporal Attack: integrates temporal motion gradients to disrupt the spatial-temporal coherence of S-HAR models.

However, current transfer-based attack on S-HAR is far from ideal due to their generally poor and unreliable performance. Recently, few studies have attempted to apply white-box S-HAR attacks against black-box models via surrogate models (Wang et al., 2021; Liu et al., 2020a). However, results show that their transfer success rate is highly determined by the specific choice of the S-HAR surrogate, so that its general adversarial transferability is low (Wang et al., 2023; Lu et al., 2023), also referred to as low/weak transferability. Although similar research in other fields (Dong et al., 2018; Huang et al., 2023; Diao et al., 2024b) has achieved success, a direct application of them on S-HAR still shows low transferability, raising doubt on the usefulness of adversarial transferability in this domain (Lu et al., 2023). More importantly, the reason for this failure remains unclear.

We begin by investigating the underlying causes of the low transferability in S-HAR attacks. By first systematically investigating the sensitivity of attack transferability on the choice of surrogates, we compare the loss surface smoothness of the surrogates, inspired by (Wu & Zhu, 2020; Qin et al., 2022). A visual comparison is shown in Figure 2, which gives a clear indication of high correlations between loss smoothness and transferability. Consequently, we argue that the transfer-based S-HAR attack should smoothen the surrogate's loss during training. Various strategies aim to achieve smoother loss landscapes, via *e.g.* regularization (Zhao et al., 2022; Foret et al., 2021) or Bayesian learning (Izmailov et al., 2018; Nguyen et al., 2024; Maddox et al., 2019). We explore the latter and use Bayesian Neural Networks (BNNs). This is because BNNs tend to have smooth loss landscapes (Blundell et al., 2015; Izmailov et al., 2018; Nguyen et al., 2024). More importantly, it enables us to attack the whole distributions of models, *i.e.* Bayesian attacks, which has been proven to enhance the transferability in other fields (Li et al., 2023; Gubri et al., 2022).

However, it is not straightforward to design such a transferable Bayesian attack for S-HAR. First, attacking a distribution of models requires sampling from the posterior distribution. But S-HAR classifiers contain at least several millions of parameters (Liu et al., 2020b), which makes sampling computationally expensive. Second, most prior transferable attacks are specifically designed for static data, e.g. images. However, most S-HAR models learn the spatial-temporal features because skeletal data contains rich motion dynamics. A naive adaptation of them ignores the spatial-temporal coherence during attack, leading to either lower transferability or excessive attack which raises suspicion. How to incorporate the motion dynamics in Bayesian attacks has not been explored.

To tackle these challenges, we propose the first **T**ransfer-based **A**ttack specifically designed for **S**keletal **A**ction **R**ecognition, TASAR, with key novelties shown in Figure 1. First, our post-train Bayesian strategy keeps a pre-trained surrogate intact by appending lightweight Bayesian components behind it, without the need for re-training of the pre-trained surrogate. Second, we propose

a novel dual Bayesian optimization for smoothed posterior sampling, which effectively smoothens the rugged loss surface. Finally, unlike previous transfer-based attacks that treat each frame independently, overlooking the temporal dependencies between sequences, we integrate the temporal motion gradient in a Bayesian manner to disrupt the spatial-temporal coherence of S-HAR models. For exhaustive evaluation, we build the first comprehensive robust S-HAR evaluation benchmark *RobustBenchHAR*. *RobustBenchHAR* consists of 7 S-HAR models with diverse GCN structures and latest Transformer structures, 10 attack methods, 3 datasets and 2 defense methods. Extensive experiments demonstrate the superiority and generalizability of TASAR.

## 2 RELATED WORK

**Skeleton-Based Human Action Recognition.** Early S-HAR research employed convolutional neural networks (CNNs) (Ali et al., 2023) and recurrent neural networks (RNNs) (Du et al., 2015) to extract motion features in the spatial and temporal domains. However, skeleton data as a topological graph challenges feature representation with traditional methods. Recent advances with graph convolutional networks (GCNs) (Kipf & Welling, 2016) have improved performance by modeling skeletons as topological graphs, with nodes corresponding to joints and edges to bones (Yan et al., 2018). Subsequent improvements in graph designs and network architectures include two-stream adaptive GCN (2s-AGCN) (Shi et al., 2019a), directed acyclic GCN (DGNN) (Shi et al., 2019b), multi-scale GCN (MS-G3D) (Liu et al., 2020b), channel-wise topology refinement (CTR-GCN) (Chen et al., 2021) and auxiliary feature refinement (FR-HEAD) (Zhou et al., 2023). Alongside advancements in GCN-based models, recent studies have explored temporal Transformer structures for S-HARs (Do & Kim, 2024; Qiu et al., 2022; Guo et al., 2024), but their vulnerability remains unexplored. Recently, robust S-HAR against adversarial noise has gained attention, with works such as Diao et al. (2024a) exploring adversarial sample distributions and Tanaka et al. (2024) applying Fourier analysis. BEAT (Wang et al., 2023) employs a post-train Bayesian strategy to achieve full Bayesian treatment on clean data, adversarial distribution and classifier. Although post-train Bayesian strategy is suggested to be more robust (Wang et al., 2023), its application in S-HAR attacks has not been explored. To address this, we introduce a new post-train Dual Bayesian strategy to improve adversarial transferability.

**Adversarial Attacks on S-HAR.** Adversarial attacks (Szegedy et al., 2013) have been applied to various data types, with increasing focus on S-HAR. CIASA (Liu et al., 2020a) proposes a constrained iterative attack via GAN (Goodfellow et al., 2014a) to regularize the adversarial skeletons, while SMART (Wang et al., 2021) uses a perception loss gradient. Tanaka et al. (2022) suggests only perturbing skeletal lengths. These methods are all white-box attacks, requiring full knowledge of the victim model. Different from existing white-box attacks leverage dynamics or physical constraints to preserve visual naturalness within white-box settings, we focus on disrupting spatial-temporal coherence to improve adversarial transferability. In contrast, BASAR (Diao et al., 2021; 2024a) proposes motion manifold searching to achieve the query-based black-box attack. FGDA-GS (Kang et al., 2023a) estimates gradient signs to further reduce query numbers. Compared to white-box and query-based attacks, transfer-based attacks (Liu et al., 2016) pose a more practical threat as real-world HAR scenarios typically cannot access white-box information or extensive querying. While existing white-box S-HAR attacks (Wang et al., 2021; Liu et al., 2020a) can be adapted for transfer-based scenarios, they suffer from low transferability and sensitivity to surrogate choices. Lu et al. (2023) proposes a no-box attack for S-HAR, but it also fails in transfer-based attacks. Various type of transfer-based attacks, including gradient-based (Dong et al., 2018; Ma et al., 2023; Ge et al., 2023), input transformation (Xie et al., 2019; Zhu et al., 2024; Wang et al., 2024), and ensemble-based methods (Xiong et al., 2022; Li et al., 2023; Tang et al., 2024), exhibit high transferability across various tasks but struggle in skeletal data (Lu et al., 2023). Therefore, there is an urgent need to develop a transferable attack for skeleton-based action recognition.

## 3 METHODOLOGY

### 3.1 PRELIMINARIES

We denote a clean motion $\mathbf{x} \in \mathcal{X}$ and its corresponding label $y \in \mathcal{Y}$. Given a surrogate action recognizer $f_\theta$ parametrized by $\theta$, $f_\theta$ is trained to map a motion $\mathbf{x}$ to a predictive distribution $p(y \mid$

$\mathbf{x}, \theta$). The white-box attack aims to find adversarial examples $\tilde{\mathbf{x}}$ within the neighborhood $\mathcal{B}_\epsilon(\mathbf{x}) = \{\tilde{\mathbf{x}} : \|\tilde{\mathbf{x}} - \mathbf{x}\|_p \leq \epsilon\}$ that misleads the target model $f_\theta$:

$$\underset{\|\tilde{\mathbf{x}} - \mathbf{x}\|_p \leq \epsilon}{\arg\min} \; p(y \mid \tilde{\mathbf{x}}, \theta), \tag{1}$$

where $\epsilon$ is the perturbation budget. $\|\cdot\|_p$ is the $l_p$ norm distance. The procedure of transfer-based attack is firstly crafting the adversarial example $\tilde{\mathbf{x}}$ by attacking the surrogate model, then transferring $\tilde{\mathbf{x}}$ to attack the unseen target model. In Equation (1), since the transferable adversarial examples are optimized against one surrogate model, the adversarial transferability heavily relies on the surrogate model learning a classification boundary similar to that of the unknown target model. While possible for image classification, it proves unrealistic for S-HAR (Wang et al., 2023; Lu et al., 2023).

## 3.2 MOTIVATION

Existing S-HAR attacks have shown outstanding white-box attack performance but exhibit low transferability (Wang et al., 2023). Similarly, previous transfer-based attacks (Dong et al., 2018; Xiong et al., 2022), successful on image data, also show poor transferability when applied to skeletal motion (Lu et al., 2023). Naturally, two questions occur to us: *(1) Why do existing adversarial attacks fail to exhibit transferability in skeletal data? (2) Do transferable adversarial examples truly exist in S-HAR?*

To answer these questions, we start by generating adversarial examples using various surrogate skeletal recognizers and then evaluate their adversarial transferability. Obviously, in Table 1, the transferability is highly sensitive to the chosen surrogates, e.g. CTR-GCN (Chen et al., 2021) as the surrogate exhibits higher transferability than ST-GCN (Yan et al., 2018). This observation motivates us to further investigate the differences between surrogate models. Previous research (Wu & Zhu, 2020; Qin et al., 2022) has proven that adversarial examples generated by surrogate models with a less smooth loss landscape are unlikely to transfer across models. Therefore, we investigate the smoothness of the loss landscape across different surrogate models. In Figure 2, we visualize the loss landscape of ST-GCN and CTR-GCN trained on the skeletal dataset NTU-60 (Shahroudy et al., 2016), and compare their smoothness to the ResNet-18 (He et al., 2016) trained on CIFAR-10 (Krizhevsky et al., 2009). More landscape visualizations can be found in Appendix C. By analyzing the loss surface smoothness, we have two findings: (1) The loss surface of models trained on skeletal data is much sharper than those trained on image data, leading to a relatively low transferability. This suggests that adversarial examples within a sharp local region are less likely to transfer across models in S-HAR, potentially explaining our first question. (2) CTR-GCN has a flatter loss landscape compared to ST-GCN, making it a more effective surrogate for higher transferability. Consequently, we argue that using a surrogate with a smoothed loss landscape will significantly enhance adversarial transferability in S-HAR.

In this work, motivated by evidence that Bayesian neural networks (BNNs) exhibit low sharpness and good generalization (Blundell et al., 2015; Maddox et al., 2019), we aim to construct a Bayesian surrogate by sampling from the model posterior space to smoothen the rugged loss landscape. From a Bayesian perspective, Equation (1) can be reformulated by approximately minimizing the Bayesian posterior predictive distribution:

$$\underset{\|\tilde{\mathbf{x}} - \mathbf{x}\|_p \leq \epsilon}{\arg\min} \; p(y \mid \tilde{\mathbf{x}}, \mathcal{D}) = \underset{\|\tilde{\mathbf{x}} - \mathbf{x}\|_p \leq \epsilon}{\arg\min} \; \mathbb{E}_{\theta \sim p(\theta|\mathcal{D})} p(y \mid \tilde{\mathbf{x}}, \theta), \tag{2}$$

where $p(\theta \mid \mathcal{D}) \propto p(\mathcal{D} \mid \theta) p(\theta)$, in which $\mathcal{D}$ is the dataset and $p(\theta)$ is the prior of model weights.

## 3.3 A POST-TRAIN BAYESIAN PERSPECTIVE ON ATTACK

Unfortunately, directly sampling from the posterior distribution of skeletal classifiers is not a straightforward task due to several factors. First, directly sampling the posterior is intractable for large-scale skeletal classifiers. Although approximate methods such as MCMC sampling (Welling & Teh, 2011) or variational inference (Blei et al., 2017) are possible, sampling is prohibitively slow and resource-intensive due to the high dimensionality of the sampling space, which typically involves at least several million parameters in skeletal classifiers. In addition, skeletal classifiers normally contain a large number of parameters and are pre-trained on large-scale datasets (Liu et al., 2019).

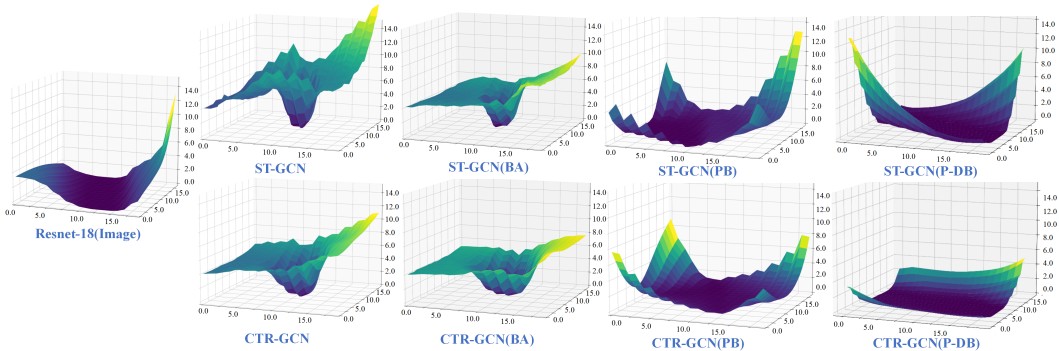

Figure 2: Comparison of loss landscapes of trained models.The $x$ and $y$ axis represent two random direction vectors sampled from a Gaussian distribution, which are added to the model's parameter space along these directions. These random direction vectors are used to assess the sensitivity of the model's loss function. The $z$ axis represents the loss value. More details can be found in Li et al. (2018). BA means the Bayesian Attack proposed by Li et al. (2023). PB means the post-train Bayesian optimization, and P-DB means the improved post-train Dual Bayesian optimization. The loss landscape optimized by post-train Dual Bayesian is significantly smoother than those of vanilla post-train Bayesian and baseline methods. More visualizations can be found in Appendix C.

Consequently, it is not practical for end-users to re-train the surrogate in a Bayesian manner, as the training process is time-consuming.

To solve the above issues, we propose a new *post-train* Bayesian attack. We maintain the integrity of the pre-trained surrogate while appending a tiny MLP layer $g_{\theta'}$ behind it, connected via a skip connection. Specifically, the final output logits can be computed as: $\text{logits} = g_{\theta'}(f_\theta(\mathbf{x})) + f_\theta(\mathbf{x})$. In practice, we adopt Monte Carlo sampling to optimize the appended Bayesian model:

$$\max_{\theta'} \mathbb{E}_{\theta' \sim p(\theta' | \mathcal{D}, \theta)} p\left(y \mid \mathbf{x}, \theta, \theta'\right) \approx \max_{\theta'_k} \frac{1}{K} \sum_{k=1}^{K} p\left(y \mid \mathbf{x}, \theta, \theta'_k\right), \theta'_k \sim p(\theta' \mid \mathcal{D}, \theta), \tag{3}$$

where $K$ is the number of appended models. Directly training such a Bayesian component is intractable, so the posterior distribution $p(\theta' \mid \mathcal{D}, \theta)$ needs to be approximated through sampling, where $p(\theta' \mid \mathcal{D}, \theta) \propto p(\mathcal{D} \mid \theta, \theta')p(\theta')$ and $p(\theta')$ is the prior of appended model weights. Correspondingly, Equation (2) can be approximately solved by performing attacks on the ensemble of tiny appended models:

$$\underset{\|\delta\|_p \leq \epsilon}{\arg\min} \frac{1}{K} \sum_{k=1}^{K} p\left(y \mid \tilde{\mathbf{x}}, \theta, \theta'_k\right), \theta'_k \sim p(\theta' \mid \mathcal{D}, \theta). \tag{4}$$

Our post-train Bayesian attack offers two advantages. First, the appended models are composed of tiny MLP layers, getting a similar memory cost to a single surrogate. Second, by freezing $f_\theta$, our post-train Bayesian strategy keeps the pre-trained surrogate intact, avoiding re-training the pre-trained surrogate. More importantly, training on $g_{\theta'}$ is much faster than on $f_\theta$ due to the smaller model size of $g_{\theta'}$.

## 3.4 POST-TRAIN DUAL BAYESIAN MOTION ATTACK

In our preliminary experiments, we found that a naive application of post-train Bayesian attack (Equation (4)) already surpassed the adversarial transfer performance of existing S-HAR attacks, which demonstrates the effectiveness of smoothening the loss surface of surrogates. However, its performance remains slightly inferior to the Bayesian attack via re-training a Bayesian surrogate (Li et al., 2023)(Equation (2)). This performance gap is understandable, as we avoid the prohibitively slow process of sampling the original posterior distribution $\theta \sim p(\theta \mid \mathcal{D})$ by using a tiny Bayesian component for post-training instead. To further eliminate the trade-off between attack strength and

efficiency, we propose a novel post-train dual Bayesian optimization for smoothed posterior sampling, to sample the appended models with high smoothness for better transferability (Figure 2). Moreover, unlike previous transfer-based attacks that assume each frame is independent and ignore the temporal dependency between sequences, we integrate motion dynamics information into the Bayesian attack gradient to disrupt the spatial-temporal coherence of S-HAR models. We name our method Post-train Dual Bayesian Motion Attack.

### 3.4.1 POST-TRAIN DUAL BAYESIAN OPTIMIZATION

This motivation is based on the view that models sampled from a smooth posterior, along with the optimal approximate posterior estimating this smooth posterior, have better smoothness (Nguyen et al., 2024). To this end, we aim for proposing a smooth posterior for learning post-train BNNs, hence possibly possessing higher adversarial transferability. Specifically, inspired by the observation that randomized weights often achieve smoothed weights update (Izmailov et al., 2018; Dziugaite & Roy, 2017; Jin et al., 2023), we add Gaussian noise to smooth the appended network weights. This is achieved by a new post-train dual Bayesian optimization:

$$\max_{\theta'} \mathbb{E}_{\theta' \sim p(\theta'|\mathcal{D},\theta)} \mathbb{E}_{\Delta\theta' \sim \mathcal{N}(\mathbf{0},\sigma^2\mathbf{I})} p\left(y \mid \mathbf{x}, \theta, \theta' + \Delta\theta'\right). \tag{5}$$

For any appended model sampled from the posterior, Equation (5) ensures that the neighborhood around the model parameters has uniformly low loss. We further use dual Monte Carlo sampling to approximate Equation (5):

$$\min_{\theta'_k \sim p(\theta'|\mathcal{D},\theta)} \frac{1}{MK} \sum_{k=1}^{K} \sum_{m=1}^{M} L\left(\mathbf{x}, y, \theta, \theta'_k + \Delta\theta'_{km}\right), \Delta\theta'_{km} \sim \mathcal{N}\left(\mathbf{0}, \sigma^2\mathbf{I}\right), \tag{6}$$

where $L$ is the classification loss. Considering dual MCMC samplings computationally intensive, we instead consider the worst-case parameters from the posterior, followed by Li et al. (2023). Hence Equation (6) can be equivalent to a min-max optimization problem, written as:

$$\min_{\theta'_k \sim p(\theta'|\mathcal{D},\theta)} \max_{\Delta\theta' \sim \mathcal{N}(\mathbf{0},\sigma^2\mathbf{I})} \frac{1}{K} \sum_{k=1}^{K} L\left(\mathbf{x}, y, \theta, \theta'_k + \Delta\theta'\right), p(\Delta\theta') \geq \xi. \tag{7}$$

The confidence region of the Gaussian posterior is regulated by $\xi$. We discuss the sensitivity to $\xi$ in the Appendix C. The entanglement between $\theta'$ and $\Delta\theta'$ complicates gradient updating. To simplify this issue, we utilize Taylor expansion at $\theta'$ to decompose the two components:

$$\min_{\theta'_k \sim p(\theta'|\mathcal{D},\theta)} \max_{\Delta\theta' \sim \mathcal{N}(\mathbf{0},\sigma^2\mathbf{I})} \frac{1}{K} \sum_{k=1}^{K} [L\left(\mathbf{x}, y, \theta, \theta'_k\right) + \nabla_{\theta'_k} L\left(\mathbf{x}, y, \theta, \theta'_k\right)^T \Delta\theta'], p(\Delta\theta') \geq \xi. \tag{8}$$

Since $\Delta\theta'$ is sampled from a zero-mean isotropic Gaussian distribution, the inner maximization can be solved analytically. We introduce the inference details, mathematical deduction and algorithm in Appendix B. As shown in Figure 2, the loss landscape optimized by post-train Dual Bayesian is significantly smoother than vanilla post-train Bayesian.

### 3.4.2 TEMPORAL MOTION GRADIENT IN BAYESIAN ATTACK

Post-train Dual Bayesian Motion Attack can be performed with gradient-based methods such as FGSM (Goodfellow et al., 2014b):

$$\tilde{\mathbf{x}} = \mathbf{x} + \alpha \cdot \text{sign}(\sum_{k=1}^{K} \sum_{m=1}^{M} \nabla L\left(\mathbf{x}, y, \theta, \theta'_k + \Delta\theta'_{km}\right)), \tag{9}$$

where $\alpha$ is the attack step size. Meanwhile, for notational simplicity, we notate the classification loss $L\left(\mathbf{x}, y, \theta, \theta'_k + \Delta\theta'_{km}\right)$ as $L\left(\mathbf{x}\right)$. Assume a motion with $t$ frames $\mathbf{x} = [x_1, x_2, \cdots, x_t]$, this attack

gradient consists of a set of partial derivatives over all frames $\nabla L(\mathbf{x}) = \left[ \frac{\partial L(\mathbf{x})}{\partial x_1}, \frac{\partial L(\mathbf{x})}{\partial x_2}, \cdots, \frac{\partial L(\mathbf{x})}{\partial x_t} \right]$. The partial derivative $\frac{\partial L(\mathbf{x})}{\partial x_t}$ assumes each frame is independent, ignoring the dependency between frames over time. This assumption is reasonable for attacks on static data such as PGD (Madry et al., 2017) while infeasible for skeletal motion attacks. In skeletal motion, most S-HAR models learn the spatial-temporal features (Yan et al., 2018), hence considering motion dynamics in the computing of attack gradient can disrupt the spatial-temporal coherence of these features, leading to more general transferability. To fully represent the motion dynamics, *first-order* (velocity) gradient $(\nabla L(\mathbf{x}))_{d1}$ and *second-order* (acceleration) gradient information $(\nabla L(\mathbf{x}))_{d2}$ should also be considered. To this end, we augment the original *position* gradient with the motion gradient, then Equation (4) becomes:

$$\tilde{\mathbf{x}} = \mathbf{x} + \alpha \cdot \text{sign}(\sum_{k=1}^{K} \sum_{m=1}^{M} \sum_{n=0}^{2} w_n (\nabla L(\mathbf{x}))_{dn}), \sum_{n=0}^{2} w_n = 1, \tag{10}$$

where $(\nabla L(\mathbf{x}))_{d0} = \nabla L(\mathbf{x})$. Motion gradient can be computed by explicit modeling (Xia et al., 2015) or implicit learning (Tang et al., 2022). Given that implicit learning requires training an additional data-driven model to learn the motion manifold, which increases computational overhead, we opt for explicit modeling. Inspired by Lu et al. (2023), we employ time-varying autoregressive models (TV-AR)(Bringmann et al., 2017) because TV-AR can effectively estimate the dynamics of skeleton sequences by modeling the temporary non-stationary signals (Xia et al., 2015). We first use first-order TV-AR($f_{d1}$) and second-order TV-AR($f_{d2}$) to model human motions respectively:

$$f_{d1} : \tilde{x}_t^i = A_t \cdot \tilde{x}_{t-1}^i + B_t + \gamma_t, \tag{11}$$

$$f_{d2} : \tilde{x}_t^i = C_t \cdot \tilde{x}_{t-1}^i + D_t \cdot \tilde{x}_{t-2}^i + E_t + \gamma_t, \tag{12}$$

where the model parameters $\beta_t^1 = [A_t, B_t]$ and $\beta_t^2 = [C_t, D_t, E_t]$ are all time-varying parameters and determined by data-fitting. $\gamma_t$ is a time-dependent white noise representing the dynamics of stochasticity. Using Equation (11), the first-order motion gradient can be derived as:

$$\left( \frac{\partial L(\tilde{\mathbf{x}}^i)}{\partial \tilde{x}_{t-1}^i} \right)_{d1} = \frac{\partial L(\tilde{\mathbf{x}}^i)}{\partial \tilde{x}_{t-1}^i} + \frac{\partial L(\tilde{\mathbf{x}}^i)}{\partial \tilde{x}_t^i} \cdot A_t. \tag{13}$$

Similarly, second-order dynamics can be expressed as below by using Equation (12):

$$\left( \frac{\partial L(\tilde{\mathbf{x}}^i)}{\partial \tilde{x}_{t-2}^i} \right)_{d2} = \frac{\partial L(\tilde{\mathbf{x}}^i)}{\partial \tilde{x}_{t-2}^i} + \frac{\partial L(\tilde{\mathbf{x}}^i)}{\partial \tilde{x}_{t-1}^i} \cdot C_{t-1} + \frac{\partial L(\tilde{\mathbf{x}}^i)}{\partial \tilde{x}_t^i} \cdot (D_t + C_t \cdot C_{t-1}), \tag{14}$$

where $C_t = \frac{\partial \tilde{x}_t^i}{\partial \tilde{x}_{t-1}^i}$ and $D_t = \frac{\partial \tilde{x}_t^i}{\partial \tilde{x}_{t-2}^i}$. After computing $\tilde{x}_{t-1}^i = C_{t-1} \cdot \tilde{x}_{t-2}^i + D_{t-1} \cdot \tilde{x}_{t-3}^i + E_{t-1} + \gamma_{t-1}$, we can compute $C_{t-1} = \frac{\partial \tilde{x}_{t-1}^i}{\partial \tilde{x}_{t-2}^i}$. Overall, the high-order dynamics gradients over all sequences can be expressed as $(\nabla L(\mathbf{x}))_{d1} = \left[ \left( \frac{\partial L(\mathbf{x})}{\partial x_1} \right)_{d1}, \left( \frac{\partial L(\mathbf{x})}{\partial x_2} \right)_{d1}, \cdots, \left( \frac{\partial L(\mathbf{x})}{\partial x_t} \right)_{d2} \right]$ and $(\nabla L(\mathbf{x}))_{d2} = \left[ \left( \frac{\partial L(\mathbf{x})}{\partial x_1} \right)_{d2}, \left( \frac{\partial L(\mathbf{x})}{\partial x_2} \right)_{d2}, \cdots, \left( \frac{\partial L(\mathbf{x})}{\partial x_t} \right)_{d2} \right]$.

## 4 EXPERIMENTS

### 4.1 *RobustBenchHAR* SETTINGS

To our best knowledge, there is no large-scale benchmark for evaluating transfer-based S-HAR attacks. To fill this gap, we build the first large-scale benchmark for robust S-HAR evaluation, named *RobustBenchHAR*. We briefly introduce the benchmark settings here, with additional details available in Appendix D.

**(A) Datasets.** *RobustBenchHAR* incorporates three popular S-HAR datasets: NTU 60 (Shahroudy et al., 2016), NTU 120 (Liu et al., 2019) and HDM05(Müller et al., 2007). Since the classifiers do not have the same data pre-processing setting, we unify the data format following (Wang et al., 2023). For NTU 60 and NTU 120, we subsampled frames to 60. For HDM05, we segmented the data into 60-frame samples.

**(B) Evaluated Models.** We evaluate TASAR in three categories of surrogate/victim models. (1) Normally trained models: We adapt 5 commonly used GCN-based models, i.e., ST-GCN (Yan et al., 2018), MS-G3D (Liu

Table 1: The attack success rate(%) of untargeted transfer-based attacks on NTU60 and NTU120. 'Ave' was calculated as the average transfer success rate over all target models except for the surrogate.'SFormer' represents SkateFormer and MI stands for MI-FGSM.

| Surrogate | Method | Dataset: NTU60 Target Models | | | | | | Ave | Dataset: NTU120 Target Models | | | | | | Ave |
|---|---|---|---|---|---|---|---|---|---|---|---|---|---|---|---|
| | | STGCN | 2sAGCN | MSG3D | CTRGCN | FRHEAD | SFormer | | STGCN | 2sAGCN | MSG3D | CTRGCN | FRHEAD | SFormer | |
| STGCN | IFGSM | 99.26 | 11.76 | 8.33 | 14.22 | 16.42 | 15.44 | 13.23 | 96.81 | 8.82 | 7.10 | 13.97 | 16.42 | 24.75 | 14.21 |
| | MI | **100.00** | 17.76 | 27.20 | 14.95 | 26.59 | 11.76 | 19.65 | 99.63 | 18.75 | **28.18** | 15.07 | 20.22 | 23.03 | 21.05 |
| | SMART | 93.28 | 5.62 | 2.19 | 6.88 | 7.19 | 10.08 | 6.39 | 94.06 | 8.28 | 7.66 | 11.09 | 10.16 | 16.12 | 10.66 |
| | CIASA | **100.00** | 3.43 | 3.43 | 7.60 | 9.80 | 8.33 | 6.52 | **100.00** | 4.16 | 4.41 | 9.07 | 8.08 | 14.95 | 8.13 |
| | MIG | 99.50 | 25.49 | 39.60 | 19.80 | 36.50 | **18.14** | 27.91 | 98.01 | 17.45 | 23.01 | 15.22 | **23.76** | 21.53 | 20.19 |
| | DIM | 77.97 | 20.54 | 34.03 | 12.13 | 28.83 | 13.11 | 21.73 | 75.61 | 10.76 | 12.25 | 12.75 | 16.21 | 23.01 | 15.00 |
| | **TASAR** | 99.29 | **42.55** | **64.60** | **20.33** | **49.41** | 17.22 | **38.82** | 99.26 | **19.60** | 19.37 | **15.28** | 22.79 | **25.24** | **20.46** |
| MSG3D | IFGSM | 25.49 | 22.79 | **100.00** | 20.10 | 24.75 | 16.66 | 21.96 | 26.96 | 16.42 | **100.00** | 15.20 | 18.38 | 27.20 | 20.83 |
| | MI | 22.42 | 13.72 | **100.00** | 14.83 | 20.22 | 12.25 | 16.69 | 25.49 | 12.25 | **100.00** | 14.46 | 16.78 | 22.30 | 18.26 |
| | SMART | 21.66 | 8.96 | **100.00** | 12.50 | 13.54 | 12.09 | 13.75 | 31.25 | 13.96 | **100.00** | 16.04 | 17.92 | 23.38 | 20.51 |
| | CIASA | 17.40 | 5.88 | **100.00** | 11.27 | 11.51 | 11.76 | 11.56 | 22.79 | 5.88 | **100.00** | 12.50 | 12.50 | 19.11 | 14.26 |
| | MIG | 31.92 | 39.65 | **100.00** | 24.44 | 36.15 | 23.06 | 31.04 | 32.17 | 27.22 | **100.00** | 23.27 | 31.18 | 33.54 | 29.48 |
| | DIM | 28.58 | 47.27 | **100.00** | 17.82 | 35.27 | 17.69 | 29.33 | 30.94 | 38.24 | **100.00** | 19.43 | 30.19 | 29.82 | 29.72 |
| | **TASAR** | **48.87** | **51.18** | 99.61 | **41.49** | **40.14** | **23.90** | **41.11** | **41.16** | **47.28** | **100.00** | **28.83** | **40.60** | **40.37** | **39.65** |
| CTRGCN | IFGSM | 27.45 | 16.54 | 13.72 | 95.22 | 44.97 | 20.71 | 24.68 | 33.33 | 14.95 | 14.33 | 97.30 | 31.00 | 31.49 | 25.02 |
| | MI | 25.36 | 23.52 | 36.51 | 99.02 | 51.34 | 19.85 | 31.32 | 30.14 | 19.73 | 29.16 | 99.26 | 29.16 | 28.30 | 27.30 |
| | SMART | 15.00 | 5.00 | 4.69 | 99.69 | 15.31 | 9.27 | 9.85 | 19.75 | 5.84 | 4.63 | 99.60 | 9.27 | 17.13 | 11.32 |
| | CIASA | 14.70 | 4.65 | 5.88 | **99.75** | 15.93 | 9.31 | 10.09 | 30.94 | 5.88 | 4.65 | **99.75** | 10.53 | 16.91 | 11.51 |
| | MIG | 28.86 | 35.34 | 48.19 | 93.55 | 53.46 | 21.04 | 37.38 | 30.94 | 24.75 | 32.67 | 94.18 | 34.03 | 29.45 | 30.37 |
| | DIM | 23.01 | 14.97 | 15.59 | 53.16 | 34.71 | 17.51 | 21.16 | 29.51 | 19.49 | 24.87 | 62.31 | 25.37 | 23.63 | 24.57 |
| | **TASAR** | **33.76** | **52.31** | **66.74** | 97.06 | **58.32** | **21.07** | **46.44** | **33.59** | **26.22** | **33.82** | 92.89 | **35.78** | **32.84** | **32.45** |
| STFormer | IFGSM | 23.03 | 15.19 | 11.27 | 14.95 | 16.42 | 13.48 | 15.72 | 26.26 | 13.97 | 12.99 | 15.44 | 20.83 | 24.50 | 19.00 |
| | MI | 18.13 | 12.29 | 19.36 | 12.25 | 19.36 | 10.78 | 15.36 | 26.22 | 21.07 | 32.35 | 15.20 | 22.54 | 23.77 | 23.53 |
| | SMART | 21.77 | 6.04 | 6.04 | 11.29 | 10.08 | 10.88 | 11.02 | 23.79 | 9.27 | 4.43 | 9.27 | 12.90 | 21.37 | 13.51 |
| | CIASA | 18.62 | 6.37 | 5.39 | 10.54 | 10.78 | 10.78 | 10.41 | 24.01 | 10.53 | 6.61 | 11.03 | 15.19 | 22.30 | 14.95 |
| | MIG | 22.31 | 21.44 | 18.89 | 16.77 | 23.44 | 16.77 | 19.94 | 30.54 | 20.32 | 21.88 | 16.46 | 21.25 | 24.87 | 22.55 |
| | DIM | 23.39 | 33.04 | 32.67 | 15.47 | 28.71 | 14.72 | 24.67 | 29.82 | 15.84 | 14.72 | 13.99 | 19.05 | 24.50 | 19.65 |
| | **TASAR** | **26.44** | **54.32** | **42.78** | **16.35** | **37.98** | **18.38** | **32.71** | **34.61** | **34.61** | **46.63** | **19.71** | **32.21** | **26.92** | **32.45** |

et al., 2020b), CTR-GCN (Chen et al., 2021), 2s-AGCN (Shi et al., 2019a), FR-HEAD (Zhou et al., 2023), and two latest Transformer-based models SkateFormer(Do & Kim, 2024) and STTFormer(Qiu et al., 2022). To our best knowledge, this is the first work to investigate the robustness of Transformer-based S-HARs. (2) Ensemble models: an ensemble of ST-CGN, MS-G3D and DGNN (Shi et al., 2019b). (3) Defense models: We employ BEAT (Wang et al., 2023) and TRADES (Zhang et al., 2019a), which all demonstrate their robustness for skeletal classifiers.

**(C) Baselines.** We compare with state-of-the-art (SOTA) S-HAR attacks, i.e. SMART (Wang et al., 2021) and CIASA (Liu et al., 2020a). We also adopt the SOTA transfer-based attacks as baselines, including gradient-based, i.e., I-FGSM (Kurakin et al., 2018), MI-FGSM (Dong et al., 2018) and the latest MIG (Ma et al., 2023), input transformation method DIM (Xie et al., 2019), and ensemble-based/Bayesian attacks, i.e., ENS (Dong et al., 2018), SVRE (Xiong et al., 2022) and BA (Li et al., 2023). For a fair comparison, we ran 200 iterations for all attacks under $l_\infty$ norm-bounded perturbation of size 0.01. For TASAR, we use the iterative gradient attack instead of FGSM in Equation (10).

**(D) Implementation Details.** Our appended model is a simple two-layer fully-connected layer network. Unless specified otherwise, we use $K = 3$ and $M = 20$ in Equation (10) for default and explain the reason in the ablation study later. More implementation details can be found in Appendix D.

## 4.2 EVALUATION ON NORMALLY TRAINED MODELS

**Evaluation of Untargeted Attack.** As shown in in Table 1, TASAR significantly surpasses both S-HAR attacks and transfer-based attacks under the black-box settings, while maintaining comparable white-box attack performance. Specifically, TASAR achieves the highest average transfer success rate of **35.5%** across different models and datasets, surpassing SMART (Wang et al., 2021) (the SOTA S-HAR attack) and MIG (Ma et al., 2023) (the SOTA transfer-based attack) by a large margin of **23.4%** and **8.1%** respectively. Moreover, TASAR shows consistent transferability across all surrogate models, target models and datasets. These improvements break the common belief that transfer-based attacks in S-HAR suffer from low transferability and highly rely on the chosen surrogate (Lu et al., 2023).

**Evaluation of Targeted Attack.** In this section, we focus on targeted attacks under the black-box setting. Improving targeted attack transferability on S-HAR is generally more challenging than untargeted attacks. This is primarily due to the significant semantic differences between the randomly selected class and the original one. Attacking a 'running' motion to 'walking' is generally easier than to 'drinking'. This is why targeted attacks have lower success rate than untargeted attacks. However, Table 2 shows TASAR still outperforms the baseline under most scenarios. Moreover, TASAR can successfully attack the original class to a target with an obvious semantic gap without being detected by humans. The visual examples can be found in Figure 4.

Table 2: The targeted attack success rate (%) of targeted transfer-based attack on NTU60.

| Surrogate | Method | Target | | | | | Ave |
|---|---|---|---|---|---|---|---|
| | | STGCN | 2sAGCN | MSG3D | CTRGCN | FRHEAD | |
| STGCN | MI | 27.45 | 3.06 | 2.32 | 1.71 | 1.71 | 2.20 |
| | SMART | 28.02 | 1.20 | 1.81 | 1.41 | 1.81 | 1.56 |
| | TASAR | **28.79** | **6.06** | **6.06** | **8.33** | **6.82** | **6.82** |
| MSG3D | MI | 2.08 | 3.31 | 32.72 | 1.83 | 2.45 | 2.42 |
| | SMART | 0.80 | 0.60 | 44.95 | 1.01 | 1.01 | 0.86 |
| | TASAR | **9.09** | **9.09** | **57.58** | **9.85** | **9.33** | **9.34** |
| CTRGCN | MI | 3.06 | 3.30 | 2.81 | 29.53 | 4.53 | 3.43 |
| | SMART | 1.61 | 1.61 | 1.61 | **43.95** | 1.81 | 1.66 |
| | TASAR | **8.33** | **9.09** | **8.33** | 22.73 | **9.09** | **8.71** |
| 2sAGCN | MI | 1.47 | **98.61** | 1.83 | 1.83 | 1.47 | 1.65 |
| | SMART | 2.21 | 53.02 | 1.20 | 2.62 | 2.21 | 2.06 |
| | TASAR | **10.61** | 76.52 | **4.56** | **10.61** | **8.33** | **8.53** |

Table 3: The untarget attack success rate (%) against defense models on HDM05 (**top**) and NTU 60 (**bottom**).

| Surrogate | Method | TRADES | | | BEAT | | |
|---|---|---|---|---|---|---|---|
| | | STGCN | MSG3D | CTRGCN | STGCN | MSG3D | CTRGCN |
| STGCN | MI-FGSM | 3.95 | 3.75 | 3.54 | **96.45** | 22.29 | 16.45 |
| | SMART | 2.81 | 3.13 | 1.88 | 80.13 | 3.34 | 2.90 |
| | TASAR | **3.92** | **4.17** | **2.94** | 92.19 | **60.16** | **39.84** |
| MSG3D | MI-FGSM | 3.02 | 3.02 | 2.42 | 36.89 | **100.00** | 30.64 |
| | SMART | 2.50 | 3.13 | 3.13 | 6.69 | 82.36 | 4.01 |
| | TASAR | **12.26** | **10.29** | **12.25** | **59.38** | **100.00** | **58.59** |
| STGCN | MI-FGSM | 16.05 | 5.51 | 8.46 | 95.83 | 30.39 | 16.05 |
| | SMART | 12.50 | 5.78 | 9.06 | 73.95 | 4.68 | 8.28 |
| | TASAR | 12.50 | **10.22** | **12.50** | **97.98** | **52.34** | **19.53** |
| MSG3D | MI-FGSM | **23.4** | 7.59 | 13.11 | 28.06 | 97.54 | 16.54 |
| | SMART | 19.45 | 7.42 | 11.72 | 26.71 | 79.68 | 13.82 |
| | TASAR | 19.79 | **14.58** | **17.71** | **40.63** | **100.00** | **32.29** |

## 4.3 EVALUATION ON ENSEMBLE AND DEFENSE MODELS

**Evaluation on Ensemble Models.** TASAR benefits from the additional model parameters added by the appended Bayesian components. For a fair comparison, we compare it with SOTA ensemble-based methods, i.e., ENS Dong et al. (2018) and SVRE Xiong et al. (2022), and the Bayesian Attack (BA) Li et al. (2023), because they also benefit from the model size. ENS and SVRE take three models ST-GCN, MS-G3D and DGNN as an ensemble of surrogate models, while BA and TASAR only take MS-G3D as the single substitute architecture. Unlike BA re-training the surrogate into a BNN, TASAR instead appends a small Bayesian component for post-training. We choose ST-GCN, 2s-AGCN, MS-G3D, CTR-GCN, FR-HEAD as the target models, and evaluate the average white-box attack success rate (WASR), average black-box attack success(BASR) and the number of parameters in Figure 3. We can clearly see that TASAR (blue line) achieves the best attack performance under both white-box and black-box settings, with an order of magnitude smaller model size. When using MSG3D (12.78M) as the surrogate model, the Bayesian components appended by TASAR only increase 0.012M parameters of the surrogate size, resulting in a memory cost comparable to that of a single surrogate. In contrast, the Bayesian surrogate model used by BA has 15 times more parameters (255.57M) than the single surrogate.

Since both BA and TASAR are Bayesian-based attacks, we compare the smoothness of their loss landscape in Figure 2. It can be seen that both BA and TASAR exhibit the ability to smoothen the loss landscape, providing empirical evidence for the Bayesian surrogate's effectiveness in smoothening the loss surface. Further, TASAR and BA achieve the top-2 performance in transfer-based attacks, highlighting the high correlations between loss smoothness and transferability. Compared to BA, TASAR exhibits a significantly flatter loss landscape, aligning with the higher transfer success rate than BA. The key difference between BA and TASAR is that TASAR samples from a smoothed posterior, which shows the benefit of smoothed posterior sampling for improving adversarial transferability.

**Evaluation on Defense Models.** As BEAT shows high robustness against S-HAR white-box attack (Wang et al., 2023), it is also interesting to evaluate its defense performance against black-box attack. We also em-

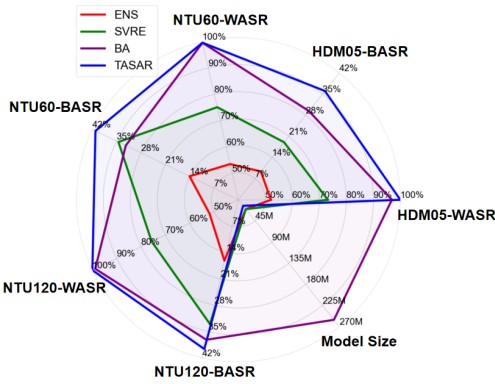

Figure 3: Comparisons with ensemble and Bayesian attacks. We calculate the model size and evaluate the average white-box (WASR) and black-box attack success rate (BASR) on the HDM05, NTU60, and NTU120 datasets, respectively.

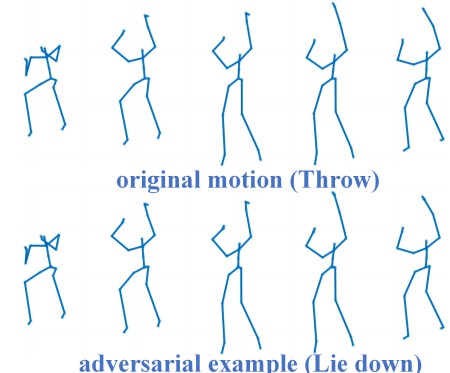

original motion (Throw)

adversarial example (Lie down)

Figure 4: The ground truth label **'Throw'** can be misclassified as **'Lie down'** on targeted attack by TASAR. The semantic differences between ground truth labels and target labels are large.

ploy the adversarial training method TRADES (Zhang et al., 2019a) as a baseline due to its robustness in S-HAR (Wang et al., 2023). Obviously, in Table 3, TASAR still achieves the highest adversarial transferability among the compared methods against defense models, further validating its effectiveness.

## 4.4  ABLATION STUDY

**Dual MCMC Sampling.** TASAR proposes a new dual MCMC sampling in the post-train Bayesian formulation (Equation (10)). To see its contribution, we conduct an ablation study on the number of appended models ($K$ and $M$ in Equation (10)). To isolate the impact of the number of appended models, we employ TASAR without motion gradient. The contribution of the motion gradient will be discussed in the subsequent ablation experiment. As shown in Table 4, compared with vanilla Post-train Bayesian strategy ($M$=0), the dual sampling significantly improves the attack performance. Furthermore, although TASAR theoretically requires intensive sampling for inference, in practice, we find a small number of sampling is sufficient ($K = 3$ and $M = 20$). More sampling will cause extra computation overhead. So we use $K = 3$ and $M = 20$ by default.

Table 4: Ablation Study on NTU 60 with ST-GCN as the surrogate. $M$ and $K$ are the number dual MCMC sampling.

| $K$ | $M$ | Target | | | | |
|---|---|---|---|---|---|---|
| | | ST-GCN | 2s-AGCN | MS-G3D | CTR-GCN | FR-HEAD |
| | 0 | 97.46 | 39.06 | 58.39 | 19.53 | 43.75 |
| 1 | 10 | 98.24 | 40.23 | 60.35 | 19.14 | 45.31 |
| | 20 | 98.05 | 41.21 | 59.57 | 18.36 | 45.72 |
| | 0 | 97.46 | 39.25 | 56.45 | 19.34 | 43.16 |
| 3 | 10 | 98.07 | 42.01 | 60.57 | 19.73 | 46.49 |
| | 20 | **99.29** | **42.55** | **64.60** | 20.33 | **49.41** |
| | 0 | 97.92 | 36.21 | 56.77 | 18.75 | 41.92 |
| 5 | 10 | 96.88 | 41.15 | 63.80 | 16.93 | 45.05 |
| | 20 | 97.14 | 39.84 | 60.94 | **20.57** | 45.21 |

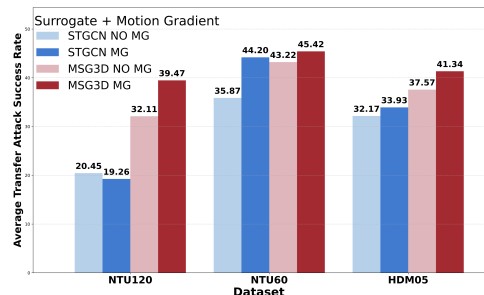

Figure 5: The ablation experiments of motion gradient. 'MG'/'No MG' means whether using motion gradient in TASAR.

**Temporal Motion Gradient.** TASAR benefits from the interplay between temporal Motion Gradient (MG) and Bayesian manner. We hence conduct ablation studies(MG/No MG) to show the effects of motion gradient and report the results in Figure 5. Compared with TASAR without using motion gradient, TASAR with motion gradient consistently improves the attack success rate in both white box and transfer-based attacks, which shows the benefit of integrating the motion gradient into the Bayesian formulation.

## 4.5  SURROGATE TRANSFERABILITY

It is widely believed that transfer-based attacks in S-HAR are highly sensitive to the surrogate choice (Lu et al., 2023; Wang et al., 2023; 2021). In this subsection, we provide a detailed analysis of the factors contributing to this phenomenon. When looking at the results in Table 1 and the visualization of loss landscape in Figure 2 and Appendix C, we note that loss surface smoothness correlates with the adversarial transferability. For example, CTR-GCN, manifesting smoother regions within the loss landscape, demonstrates higher transferability than ST-GCN and STTFormer. STTFormer trained on NTU 120 has a smoother loss surface than ST-GCN (see Appendix C), resulting in higher transferability than ST-GCN. For NTU 60, STTFormer shows a similar loss surface to that of ST-GCN and exhibits comparable transferability. Therefore, we suspect that the loss surface smoothness plays a pivotal role in boosting adversarial transferability for S-HAR, potentially outweighing the significance of gradient-based optimization techniques. Next, two-stream MS-G3D shows the highest transferability. Unlike other surrogates, which solely extract joint information, MS-G3D uses a two-stream ensemble incorporating both joint and bone features, thereby effectively capturing relative joint movements. In conclusion, we suggest that skeletal transfer-based attacks employ smoother two-stream surrogates incorporating both joint and bone information.

## 5  CONCLUSION

In this paper, we systematically investigate the adversarial transferability for S-HARs from the view of loss landscape, and propose the first transfer-based attack on skeletal action recognition, TASAR. We build *RobustBenchHAR*, the first comprehensive benchmark for robustness evaluation in S-HAR. We hope that *RobustBenchHAR* could contribute to the adversarial learning and S-HAR community by facilitating researchers to easily compare new methods with existing ones and inspiring new research from the thorough analysis of the comprehensive evaluations.

## 6 ACKNOWLEDGMENT

This project has received funding from National Natural Science Foundation of China (No. 62302139, No. 62406320), FRFCU-HFUT (JZ2023HGTA0202, JZ2023 HGQA0101).

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

## A    ETHICS AND REPRODUCIBILITY STATEMENT

**Ethics Statement.** TASAR is capable of generating natural-looking adversarial examples in S-HAR that can transfer across different skeletal classifiers. We acknowledge the possibility that TASAR might pose a significant practical threat to the current S-HAR models. However, we believe that in order to build a reliable and robust action recognizer, it is of great necessity to investigate their vulnerability. Therefore, this paper can raise awareness of vulnerability in existing S-HAR models, which greatly outweighs its risk. TASAR can be employed to evaluate the robustness of skeletal classifiers in real-world applications or improve their robustness through adversarial training.

**Reproducibility Statement.** To ensure the reproducibility of our work, we have included a comprehensive Reproducibility Statement. For the datasets used in our experiments, all the datasets used in this paper are open dataset and are available to the public. We have provided a thorough description of the data processing steps in the supplementary materials. For the novel model and algorithms presented in this work, we have included a link to the downloadable source code and model checkpoint to use our proposed benchmark and build our approach. The source code and model checkpoint can also be found in the supplementary materials. Additionally, all inference details and mathematical deduction can be found in the Appendix B. This Reproducibility Statement is intended to guide readers to the relevant resources that will aid in replicating our work, ensuring transparency and clarity throughout.

# B INFERENCE DETAILS

The detailed inference process for Post-train Dual Bayesian Motion Attack is outlined in Algorithm 1.

**Post-train Dual Bayesian Optimization.** The confidence region of the Gaussian posterior in Equation (8) regulated by $\xi$. As $\Delta\theta'$ is sampled from a zero-mean isotropic Gaussian distribution, the inner maximization can be solved analytically:

$$\Delta\theta'_* = \lambda_{\xi,\sigma}\nabla_{\theta'} L\left(\mathbf{x}, y, \theta, \theta'\right) / \left\|\nabla_{\theta'} L\left(\mathbf{x}, y, \theta, \theta'\right)\right\|. \tag{15}$$

Then the gradient of $\nabla_{\theta'_k} L\left(\mathbf{x}, y, \theta, \theta'_k\right)^T \Delta\theta'$ in Equation (8) becomes $\nabla_{\theta'_k} L\left(\mathbf{x}, y, \theta, \theta'_k\right) + \mathbf{H}\Delta\theta'_*$, in which $\mathbf{H}\Delta\theta'_*$ can be approximately estimated via the finite difference method:

$$\mathbf{H}\Delta\theta'_* \approx \frac{1}{\gamma}\left(\nabla_{\theta'_k}\frac{1}{K}\sum_{k=1}^{K} L\left(\mathbf{x}, y, \theta, \theta'_k + \gamma\Delta\theta'_*\right) - \nabla_{\theta'_k}\frac{1}{K}\sum_{i=1}^{K} L\left(\mathbf{x}, y, \theta, \theta'_k\right)\right), \tag{16}$$

where $\gamma$ is a small positive constant. Therefore, our final optimization objective is:

$$\frac{1}{K}\sum_{k=1}^{K}\nabla_{\theta'_k} L\left(\mathbf{x}, y, \theta, \theta'_k\right) + (1/\gamma)\left(\nabla_{\theta'_k} L\left(\mathbf{x}, y, \theta, \theta'_k + \gamma\Delta\theta'_*\right) - \nabla_{\theta'_k} L\left(\mathbf{x}, y, \theta, \theta'_k\right)\right). \tag{17}$$

Followed by Wang & Diao (2023), we use Stochastic Gradient Adaptive Hamiltonian Monte Carlo (SGAHMC) (Springenberg et al., 2016) for the post-train dual Bayesian optimization. Our dual Bayesian optimization assume $\theta'$ is sampled from Gaussian posterior, which has a presumed isotropic covariance matrix. In practice, we follow the suggestions from Li et al. (2023) to calculate the mean and the covariance from data by using SWAG (Maddox et al., 2019), as SWAG can offer an improved approximation to the posterior over parameters. While the posterior still relies on Gaussian approximation, it specifically incorporates the SWA(Izmailov et al., 2018) solution as its mean, and decomposes the covariance into a low-rank matrix and a diagonal matrix:

$$\theta'_k \sim \mathcal{N}\left(\theta'_{k,\text{SWA}}, \mathbf{\Sigma}_{\text{SWAG}}\right),$$
$$\mathbf{\Sigma}_{\text{SWAG}} = \frac{1}{2}\left(\mathbf{\Sigma}_{\text{diag}} + \mathbf{\Sigma}_{\text{low-rank}}\right), \tag{18}$$

where $\frac{1}{2}(\geq 0)$ can be set to other coefficients, which represent the scaling factor of SWAG for disassociating the learning rate of the covariance. Note that the posterior discussed in the preceding section is formulated based on the worst cases, thus facilitating its effortless integration with SWAG to expand diversity and flexibility. Since $\theta_{k,\text{SWA}}$ is unknown before training terminates, the dispersion of $\theta'_k$ in the final Bayesian model originates from the combination of two distinct independent Gaussian distributions, with their covariance matrices aggregated. Consequently Equation (18) becomes:

$$\theta'_k \sim \mathcal{N}\left(\theta'_{k,\text{SWA}}, \mathbf{\Sigma}_{\theta'_k}\right),$$
$$\mathbf{\Sigma}_{\theta'_k} = \alpha\left(\mathbf{\Sigma}_{\text{diag}} + \mathbf{\Sigma}_{\text{low-rank}}\right) + \beta\mathbf{I}, \tag{19}$$

where $\beta$ controls the covariance matrix of the isotropic Gaussian distribution mentioned before.

# C ADDITIONAL EXPERIMENTAL RESULTS AND ANALYSIS

## C.1 ADDITIONAL LANDSCAPE VISUALIZATIONS

To explore the sensitivity of transfer-based attacks to surrogate models (Lu et al., 2023; Wang et al., 2023; 2021), we present in Figure 6 and Figure 7 the loss landscapes of different surrogate models trained on NTU60 and NTU120. It is evident that both the post-train Bayesian optimization(PB) and the improved post-train Dual Bayesian optimization(P-DB) can smooth the loss landscape; Notably, surrogates refined by P-DB display smoother loss landscapes, leading to superior transferability over normally trained models and those optimized by PB.

---

**Algorithm 1:** Inference for Post-train Dual Bayesian Motion Attack

---

1 **Input**: $\mathbf{x}$: training data; $N_{post-train}$: the number of post-train iterations; $N_{dual}$: the number of Post-train Dual Bayesian optimization iterations; $M_{\theta'}$: sampling iterations for $\theta'_k$; $c$: moment update frequency; $K$:the number of appended models;$I$:attack iterations; $\theta$: pre-trained surrogate weights;

2 **Output**: $\{\theta'_1 + \Delta\theta'_{11}, \ldots, \theta'_K + \Delta\theta'_{KM}\}$: appended surrogate weights; $\tilde{\mathbf{x}}$: adversarial examples;

   `// Post-train Bayesian`

3 **Init**: randomly initialize $\{\theta'_1, \ldots, \theta'_K\}$;

4 **for** *i = 1 to $N_{post-train}$* **do**

5      **for** *k = 1 to $K$* **do**

6          Randomly sample a mini-batch data $\{\mathbf{x}, y\}$;

7          $\theta'_{k_i} \leftarrow \theta'_{k_{i-1}} - \eta\nabla_{\theta'_{k_i}} L\left(\mathbf{x}, y, \theta, \theta'_k\right)$;

8          **for** *t = 1 to $M_{\theta'}$* **do**

9              Update $\theta'_k$ via SGAHMC;

10          **end**

11      **end**

12 **end**

   `// Post-train Dual Bayesian Optimization`

13 each $\overline{\theta'_k} \leftarrow \theta'_{k_0}, \overline{\theta'_k}^2 \leftarrow {\theta'_{k_0}}^2$;

14 **for** *i = 1 to $N_{dual}$* **do**

15      **for** *k = 1 to $K$* **do**

16          Randomly sample a mini-batch data $\{\mathbf{x}, y\}$;

17          $\theta'_{k_i} \leftarrow \theta'_{k_{i-1}} - \eta\nabla_{\theta'_{k_i}} L\left(\mathbf{x}, y, \theta, \theta'_k\right)$;

18          Compute $\Delta\theta'_*$ via Equation (15);

19          Solving outer min optimization in Equation (8) via Equation (17);

20          **if** $\text{MOD}(i, c) = 0$ **then**

21              $n \leftarrow i/c$ ;

22              $\overline{\theta'_k} \leftarrow \frac{n\overline{\theta'_k} + \theta'_{k_i}}{n+1}, \overline{\theta'_k}^2 \leftarrow \frac{n\overline{\theta'_k}^2 + {\theta'_{k_i}}^2}{n+1}$;

23          **end**

24      **end**

25 **end**

26 $\theta'_{k,\text{SWA}} = \overline{\theta'_k}, \quad \Sigma_{diag} = \overline{\theta'_k}^2 - \overline{\theta'_k}^2$;

   `// Attack`

27 Initialization:$\tilde{\mathbf{x}}^0 = \mathbf{x}$;

28 obtain the time-varying parameters with TV-AR;

29 **for** *i = 1 to I-1* **do**

30      models=[] **for** *k = 1 to $K$* **do**

31          **for** $m = 1$ *to $M$* **do**

32              Draw $\theta'_k + \Delta\theta'_{km}$ in Equation (19);

33              models.append($\theta'_k + \Delta\theta'_{km}$);

34          **end**

35      **end**

36      Obtain the ensemble gradient;

37      Calculate the motion gradient $\left(\frac{\partial L(\tilde{\mathbf{x}}^i)}{\partial \tilde{x}^i_{t-1}}\right)_{d1}$ and $\left(\frac{\partial L(\tilde{\mathbf{x}}^i)}{\partial \tilde{x}^i_{t-2}}\right)_{d2}$ with Equation (13) and Equation (14);

38      Update $\tilde{\mathbf{x}}^{i+1}$ via Equation (10);

39 **end**

40 **return** $\{\theta'_1 + \Delta\theta'_{11}, \ldots, \theta'_K + \Delta\theta'_{KM}\}, \tilde{\mathbf{x}}$;

---

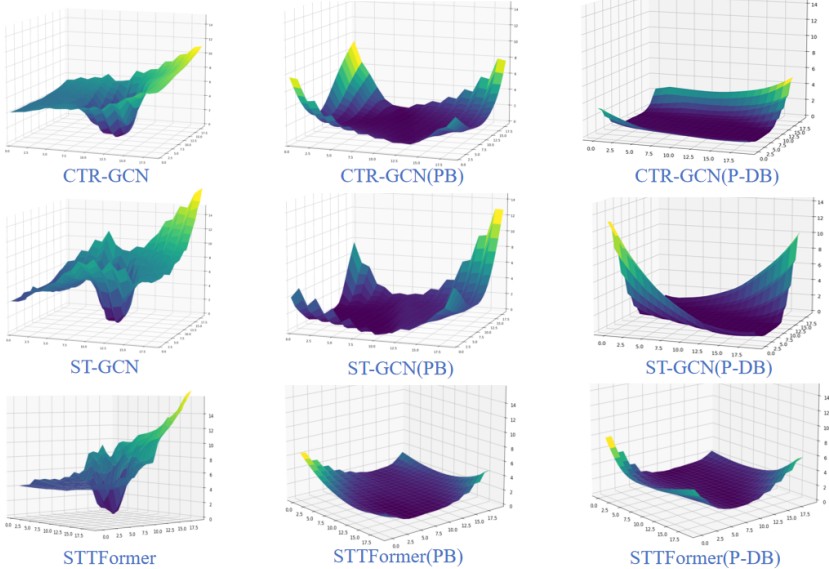

Figure 6: Loss landscapes of trained models with different methods on NTU60. The loss landscape in each plot are generated from the same original data randomly selected from the test dataset of NTU60.PB means the post-train Bayesian optimization, P-DB means the improved post-train Dual Bayesian optimization. The first row, second row and third row represent the loss surface of CTR-GCN, ST-GCN and STTFormer trained normally, as well as PB and P-DB, respectively, with the $z$ axis ranging from 0 to 16.

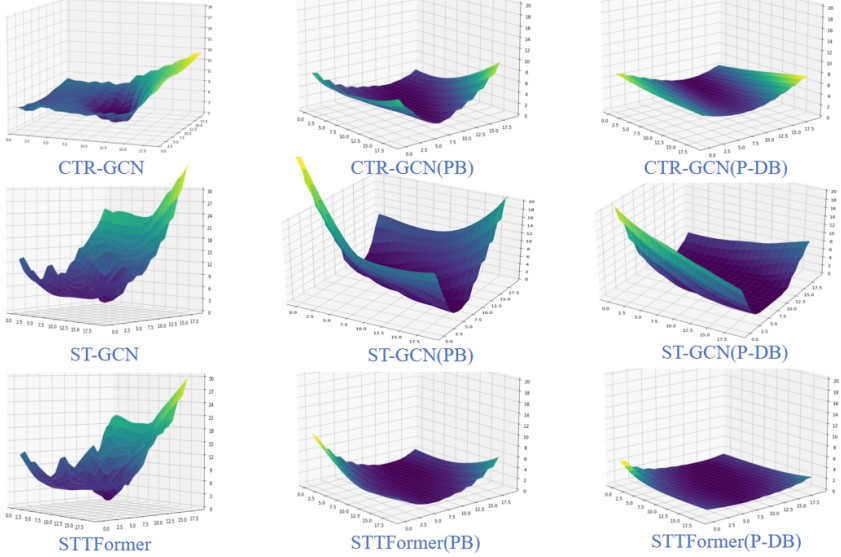

Figure 7: Loss landscapes of trained models with different methods on NTU120.The loss landscape in each plot are generated from the same original data randomly selected from the test dataset of NTU120. PB means the post-train Bayesian optimization, P-DB means the improved post-train Dual Bayesian optimization.The first row, second row, and third row correspond to the loss surfaces of CTR-GCN, ST-GCN, and STTFormer under normal training, PB and P-DB, respectively. For the normally trained plots, the $z$ axis ranges from 0 to 30, while for PB and P-DB, the range is from 0 to 20.

## C.2 Additional Results

**The Sensitivity to $\xi$.** In our Post-train Dual Bayesian optimization, we consider the worst-case parameters from the posterior. The confidence region of the Gaussian posterior is regulated by $p(\Delta\theta') \geq \xi$, influencing the extent of exploration within the posterior distribution. Therefore, we investigate the relationship between the sensitivity of $\xi$ and the performance of TASAR. Based on our assumption of an isotropic Gaussian distribution, we got the analytical solution of the worst case as below:

$$\Delta\theta'_* = \lambda_{\xi,\sigma} \nabla_{\theta'} L\left(\mathbf{x}, y, \theta, \theta'\right) / \left\|\nabla_{\theta'} L\left(\mathbf{x}, y, \theta, \theta'\right)\right\|. \tag{20}$$

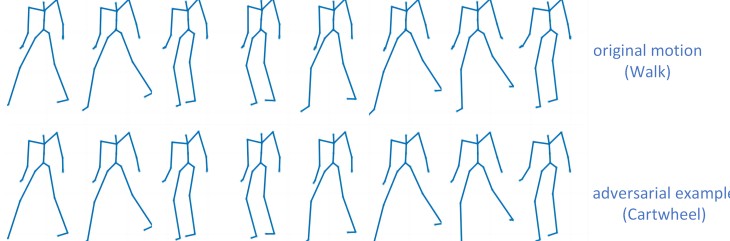

original motion
(Walk)

adversarial example
(Cartwheel)

Figure 8: The ground truth label **'Walk'** can be misclassified as **'Cartwheel'** on targeted attack by TASAR. The semantic differences between ground truth labels and target labels are large.

In Equation (20), $\xi$ can be reparameterized as a hyper-parameter $\lambda_{\xi,\sigma}$. Consequently, we conduct ablation studies to investigate the relationship between the performance of our method and the sensitivity of the hyper-parameter $\lambda_{\xi,\sigma}$. We varied with $\lambda_{\varepsilon,\sigma} \in \{0.001, 0.01, 0.05, 0.1, 1, 1.5, 2\}$ on NTU120 dataset and show the success rates of attacking victim models. We choose ST-GCN as the pre-trained model and the results are shown in Table 5. We found that setting a small value of $\lambda_{\varepsilon,\sigma}$ achieves the best adversarial transferability while maintaining a high benign accuracy. Hence we sample a collection of new surrogates near to the original surrogates and set $\lambda_{\varepsilon,\sigma} = 0.001$ as default.

Table 5: Comparsions attack success rate(%) with different $\lambda_{\xi,\sigma}$. The surrogate model is uniformly selected as ST-GCN on NTU120.

| $\lambda_{\xi,\sigma}$ | Target | | | | | Accuracy |
|---|---|---|---|---|---|---|
| | ST-GCN | 2s-AGCN | MS-G3D | CTR-GCN | FR-HEAD | |
| 0.001 | **99.26** | **19.60** | **19.37** | 15.28 | 22.79 | 63.60 |
| 0.01 | 95.88 | 17.62 | 13.10 | 14.29 | 20.23 | **64.30** |
| 0.05 | 96.43 | 17.26 | 13.10 | 14.88 | 21.43 | 63.60 |
| 0.1 | 96.43 | 17.62 | 13.10 | 13.69 | 21.43 | 60.56 |
| 1 | 96.43 | 18.45 | 11.90 | 16.07 | 23.21 | 60.39 |
| 1.5 | 96.43 | 18.45 | 13.10 | 15.48 | **23.81** | 56.58 |
| 2 | 97.02 | 17.26 | 13.10 | **16.67** | 22.02 | 54.61 |

**Perturbation Budget** In this section, we analyze the impact of attack strength on adversarial transferability. We increase the perturbation budget from 0.01 to 0.05, the results are shown in Figure 9. The general pattern of the lines aligns with the threshold setting, indicating that a larger perturbation budget consistently enhances adversarial transferability across various surrogate models.

**The Visual Quality of Targeted Adversarial Examples** TASAR can successfully attack the original class to a target with an obvious semantic gap without being detected by humans. We show an additional visual example in Figure 8.

**Comparsion with Different Training Strategies.** We conducted additional experiments to compare the performance of modeling $p\left(\theta' \mid D, \theta\right)$, $p(\theta \mid D)$ and $p\left(\theta, \theta' \mid D\right)$. The default setting of TASAR corresponds to $p\left(\theta' \mid D, \theta\right)$, where $\theta'$ is trainable while $\theta$ remains fixed. $p(\theta \mid D)$ represents a standard Bayesian neural network without adding appended models. For this case, we train the Bayesian surrogates with a cyclical variant of Stochastic Gradient Markov Chain Monte Carlo[1] to sample 3 models from the posterior distribution of neural network weights. For $p\left(\theta, \theta' \mid D\right)$, 9 appended models are added behind BNNs and both $\theta$ and $\theta'$ are trainable. Due to the high optimization complexity(updating both $\theta$ and $\theta'$), we use vanilla post-train Bayesian optimization instead of improved post-train dual Bayesian optimization optimization for updating $\theta'$.

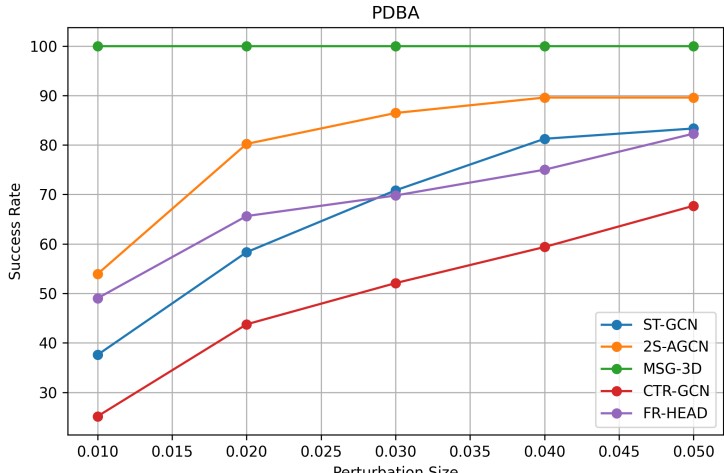

Figure 9: The success rate with different perturbation size. The surrogate model is uniformly chosen as MSG-3D and the dataset is NTU60.

Table 6: The attack success rate(%) of untargeted attacks on NTU60.

| Surrogate | Method | NTU60 | | | | | |
|---|---|---|---|---|---|---|---|
| | | ST-GCN | 2s-AGCN | MS-G3D | CTR-GCN | FR-HEAD | SFormer |
| ST-GCN | $p(\theta'|D,\theta)$ (TASAR) | **99.29** | **42.55** | 64.60 | **20.33** | **49.41** | 17.22 |
| | $p(\theta|D)$ | 90.63 | 37.11 | 69.53 | 17.58 | 43.36 | **25.39** |
| | $p(\theta,\theta'|D)$ | 93.75 | 41.80 | **73.04** | 18.75 | 46.87 | 22.65 |

The results are presented in Table 6. Compared to modeling $p(\theta \mid D)$ and $p(\theta, \theta'|D)$, TASAR achieves superior performance in transfer attacks on three out of five black-box models and demonstrates the best white-box attack performance on ST-GCN. This advantage arises from the use of improved post-training Dual Bayesian optimization, which enables smoothed posterior sampling and improves adversarial transferability. Moreover, unlike modeling $p(\theta \mid D)$ and $p(\theta, \theta'|D)$, where averaging gradients from multiple surrogates diminish white-box attack effectiveness, our post-training strategy preserves the pre-trained model intact, not reducing the original white-box attack strength. Further, TASAR significantly reducing computational overhead and accelerating training process, as shown in Table 7.

**Quantification of Model Smoothness.** We quantitatively measure the changes in loss as $\mathbf{x}$ is perturbed along a random direction with varying magnitudes. Specifically, we first sample $\mathbf{d}$ from a Gaussian distribution and normalize it onto the $\ell_2$ unit norm ball as $\mathbf{d} \leftarrow \frac{\mathbf{d}}{\|\mathbf{d}\|_F}$. Then, we calculate the loss change (smoothness) $f(a)$ for different magnitudes $a$:

$$f(a) = |L(\mathbf{x} + a \cdot \mathbf{d}, y, \theta) - L(\mathbf{x}, y, \theta)|. \tag{21}$$

We calculate $f(a)$ 20 times with different randomly sampled $\mathbf{d}$, and take the average results. For fair comparison, we use the same $\mathbf{d}$ each sampling in both 'NT' and 'TASAR'. The experimental results reveal that TASAR achieves a significantly smoother loss landscape compared to Normally Train (NT) across all magnitudes of perturbation. For large perturbations ($|a| = 1.0$), TASAR reduces the loss change by approximately threefold compared to NT. Additionally, TASAR maintains a more uniform smoothness across different magnitudes, while NT exhibits sharper variations, with larger loss changes. This indicates that TASAR effectively smoothens the loss landscape, contributing to improved transferability.

# D DETAILED *RobustBenchHAR* SETTINGS

Here we report the detailed experimental settings in our experiments. All experiments are conducted on one NVIDIA GeForce RTX 3090 GPU.

**(A) Datasets.** We choose three widely adopted datasets: NTU60 (Shahroudy et al., 2016) , NTU120 (Liu et al., 2019) and HDM05(Müller et al., 2007). The HDM05 dataset comprises 130 action classes and includes

Table 7: Model Size and Training Time measurement on NTU60. 'MS' means The Model Size(M) and 'TT' means The Training Time(hours).

| Surrogate | Method | NTU60 | |
|---|---|---|---|
| | | MS | TT |
| ST-GCN | $p(\theta'|D,\theta)$ | **3.54** | **0.65** |
| | $p(\theta|D)$ | 9.30 | 5.7 |
| | $p(\theta,\theta'|D)$ | 9.36 | 6.4 |

Table 8: Loss changes ($f(a)$) measurement for normally trained Surrogate and TASAR on HDM05. "NT" means "Normally Training".

| Surrogate | Method | Magnitude | | | | | | | |
|---|---|---|---|---|---|---|---|---|---|
| | | -1.0 | -0.8 | -0.6 | -0.4 | 0.4 | 0.6 | 0.8 | 1.0 |
| ST-GCN | NT | 7.46 | 6.05 | 4.34 | 2.08 | **2.06** | 4.33 | 5.91 | 7.24 |
| | TASAR | **2.66** | **1.70** | **1.03** | **1.77** | 2.27 | **1.40** | **1.65** | **2.31** |

2337 sequences performed by 5 non-professional actors. NTU60 offers 60 action classes, it comprises 56,880 videos captured from 40 subjects across 155 camera viewpoints. NTU120 extends NTU60 with 120 action classes, it contains 114,480 videos from 106 subjects across 155 camera viewpoints. Due to variations in data pre-processing settings among S-HAR classifiers (such as data requiring subsampling(Zhang et al., 2019b)), we unify the data format following Wang et al. (2023). For NTU60 and NTU120, we subsample frames to 60. For HDM05, we segment the data into 60-frame samples(Diao et al., 2021).

**(B) Evaluated Models.** We evaluate TASAR in three categories of surrogate/victim models.

(1) Normally trained models: We adapt 5 commonly used GCN-based models, i.e., ST-GCN(Yan et al., 2018), 2S-AGCN(Shi et al., 2019a), CTR-GCN(Chen et al., 2021), MS-G3D(Liu et al., 2020b), FR-HEAD(Zhou et al., 2023) and 2 Transformer-based models, i.e., STTFormer(Qiu et al., 2022) and SkateFormer(Do & Kim, 2024). Below we introduce the seven skeletal classifiers in details. ST-GCN (Yan et al., 2018) is the first time to apply graph convolution to S-HAR, constructing nodes and edges in the topology using skeletal information. CTR-GCN (Chen et al., 2021) improves the design of GCNs of ST-GCN and proposes to dynamically learn different topologies by learning a shared topology as a common prior for all channels and refining it for each channel. 2s-AGCN (Shi et al., 2019a) enables the model to learn graph topologies end-to-end through self-attention. It also incorporates a dual-stream framework to model first-order and second-order information simultaneously. MS-G3D (Liu et al., 2020b) proposes a disentangled multi-scale aggregation scheme to eliminate redundant dependencies between node features from different neighborhoods, thereby capturing global joint relationships on human skeletons. FR-HEAD (Zhou et al., 2023) applies contrastive feature refinement at various stages of GCNs to build multi-level feature extraction. This allows ambiguous samples to be dynamically discovered and calibrated in the feature space. STTFormer (Qiu et al., 2022) divides the skeleton sequence into temporal tuples to capture the relationships between different joints in consecutive and non-adjacent frames. SkateFormer (Do & Kim, 2024) classifies essential skeletal-temporal relationships for S-HAR into four distinct categories and utilizes self-attention in each partition to focus on key joints and frames crucial for recognition. To the best of our knowledge, this is the first work to investigate the robustness of Transformer-based S-HARs.

(2) Ensemble model: An ensemble of ST-CGN, MS-G3D and DGNN (Shi et al., 2019b).

(3) Defense models: We employ BEAT (Wang et al., 2023) and TRADES (Zhang et al., 2019a), which all demonstrate their robustness for skeletal classifiers. BEAT (Wang et al., 2023) proposes a black-box defense framework, which transforms any pre-trained classifier into a more robust one by maximizing the joint probability of clean data, adversarial examples and the classifier through joint Bayesian treatments.TRADES (Zhang et al., 2019a) is a white-box defense method that introduces a KL-divergence loss function for adversarial training. TRADES not only accounts for natural error but also incorporates adversarial error, balancing robustness and accuracy.

**(C) Baselines.** Unlike images or videos, the space available for attacking skeletons is much smaller, making adversarial perturbations on skeletons more easily detectable, here we choose two state-of-the-art attacks against S-HAR: (1) SMART (Wang et al., 2021) ensures the imperceptibility of the attack by introducing an adversarial attack perceptual loss function for S-HAR. (2) CIASA (Liu et al., 2020a)maintains the spatio-temporal constraints of joint connections and skeletal bone lengths through spatial skeleton realignment and further ensures the anthropomorphic plausibility by utilizing GAN(Goodfellow et al., 2014a) for regularization.

Besides the attacks specifically designed for S-HAR, we also select general transfer-based attacks as baselines, these attacks include (1) Gradient-based attacks, such as I-FGSM (Kurakin et al., 2018), an iterative fast gra-

dient method; MIFGSM (Dong et al., 2018), which integrates momentum into I-FGSM to stabilize the update direction and prevent getting stuck in poor local maxima; and the latest MIG (Ma et al., 2023), which uses integrated gradients to steer the generation of adversarial perturbations and adjusts them according to the momentum updating strategy. (2) Input transformation attacks, such as DIM (Xie et al., 2019), which improves the transferability of adversarial examples by creating diverse input patterns. (3) Ensemble-based/Bayesian attacks, including ENS (Dong et al., 2018), which attacks multiple models with fused logit activations; SVRE (Xiong et al., 2022), which escapes poor local optima by computing an unbiased gradient estimate through variance reduction for each iteration; and BA (Li et al., 2023), which fine-tunes the weights of the surrogate model in a Bayesian manner, thereby creating an ensemble of infinitely many DNNs as surrogates.

For a fair comparison, we run 200 iterations for all attacks under $l_\infty$ norm-bounded perturbation of size 0.01. For TASAR, we use the iterative gradient attack instead of FGSM.

**(D) Implementation Details Of TASAR.** Our appended model is a simple two-layer fully-connected layer network. Unless specified otherwise, we use $K = 3$ and $M = 20$ in TASAR for default and explain the reason in the ablation study later.

During the post-train, we set a learning rate of 0.03 with five epochs. We use *SGAHMC* optimizers (Springenberg et al., 2016), within it $\tau$ is automatically chosen, the friction coefficient $F = 10^{-5}$ and $M_{\theta'} = 30$ steps for sampling.

During the dual Bayesian optimization, we set $\gamma = 0.1/ \|\Delta\theta'_*\|_2$ and perform training for 5 epochs with a learning rate of 0.03. Additionally, we always set $\lambda_{\varepsilon,\sigma} = 0.001$.

During Inference, SWAG adjusts the covariance matrix using a constant multiplier to decouple the learning rate from covariance(Maddox et al., 2019). Here we always use 1.5 as the rescaling factor. When performing attacks, we set $\sigma = 0.009$ for models with SWAG. The $w_1$,$w_2$ and $w_3$ are set as 0.8, 0.1 and 0.1.

**(E) Computing Resource.** The experimental platform used in this study is equipped with an AMD EPYC 7542 32-Core CPU operating at a clock speed of 2039.813 GHz, four NVIDIA GeForce RTX 3090 GPUs, and 24 GB of memory per GPU. The proposed method was implemented using the open-source machine learning framework PyTorch.

