# OpenReview forum: "TASAR: Transfer-based Attack on Skeletal Action Recognition"
_ICLR.cc/2025/Conference — ICLR 2025 Poster_

### Official Review · Reviewer_WYTN · 2024-10-28

**Soundness:** 4
**Presentation:** 3
**Contribution:** 3
**Rating:** 6
**Confidence:** 4

**Summary:**

This paper investigates the issue of weak adversarial transferability in skeleton-based Human Activity Recognition (S-HAR) attacks. It proposes a new transfer-based attack called TASAR, which aims to improve transferability by smoothing the loss surface. The authors also establish a large-scale robust S-HAR benchmark for comprehensive evaluation.

**Strengths:**

The paper identifies a crucial limitation of existing S-HAR attacks and proposes a practical solution. The proposed TASAR attack is innovative and demonstrates potential improvements in adversarial transferability. The evaluation on the large-scale benchmark provides valuable insights into the effectiveness of existing and proposed methods.

The author has submitted the code, making the work reproducible.

**Weaknesses:**

I suggest that the authors cite more papers related to recent research from top conferences over the past two years.

More related literature can be included, for instance, by adding a topic on noise-robust human action recognition.

**Questions:**

N/A

---

> ### Author Response · Authors · 2024-11-24
> **Response to Reviewer WYTN**
>
> Thank you for your valuable feedback and generous support.
>
> > Q1: I suggest that the authors cite more papers related to recent research from top conferences over the past two years. More related literature can be included, for instance, by adding a topic on noise-robust human action recognition.
>
> A1: Following your suggestions, we have expanded the discussion in Sec.2 Related Work to include more studies from the past two years on transfer-based attacks and robust S-HAR approaches against adversarial noise.

---

> ### Author Response · Authors · 2024-11-27
>
> Dear Reviewer WYTN
>
> We sincerely thank you for your positive feedback on our paper! If you have any further questions or suggestions, please let us know!
>
> Best,
>
> Authors

---

### Official Review · Reviewer_LgjH · 2024-10-30

**Soundness:** 3
**Presentation:** 2
**Contribution:** 3
**Rating:** 5
**Confidence:** 3

**Summary:**

This paper explores an interesting research problem in skeleton-based action recognition.

This paper proposes the transfer-based attack on skeletal action recognition.

The proposed model incorporates motion dynamics into the Bayesian attack gradient, effectively disrupting the spatial-temporal coherence of the skeletal action recognition.

The authors explore and analyse several model architectures, and conduct experimental analysis and evaluations on several benchmarks and show that the proposed model achieves good performance compared to existing baselines.

**Strengths:**

+ Overall this paper explores an interesting research problem.

+ The authors conduct large-scale evaluations to show that the proposed model achieves good performance on several benchmarks.

+ Some of the figures presented in the paper are quite nice and provide some insights to readers.

**Weaknesses:**

Major:

- One of the major issues of this paper is that some of the concepts presented in the paper are not properly and clearly explained. For example, in abstract section, what is "white-box scenarios". Also what does it mean by saying "its sharpness contributes to the weak transferability in S-HAR"?

- Fig. 1 is not being introduced and explained clearly. For example, what is spatial attack, spatial-temporal attack, why frozen, trainable, being lightweight and demanding? What are appended models and dual bayesian? These concepts should be clearly explained to readers, either in texts or figure caption. Moreover, how to tell which one is better?

- The introduction section is not very well-motivated. For example, in Line 83, referring to Table 1; however, the concepts to understand the table are not being explained. The authors are encouraged to reorganise the section, explain the concepts properly before going deep into the technical jargon. What is "white-box"? In contribution section, what is "low adversarial transferability"?

- In related work section, the authors outline some closely related works; however, they do not highlight what are the major differences between existing works and the proposed method, what are the insights, how the proposed method has the potential to address the challenges etc.

- Fig. 2, the font sizes are invisible to reviewer, what are the x and y axis represent for? What do "random direction vectors" mean? These concepts should be properly explained.

- In Eq. (2), what are boldface $x$ and $\tilde{x}$ mean?  In Line 158 and 183, it says "a clean motion" and "the adversarial sample", what do they mean and how they align with each other, what would be the relationship between them? These concepts and maths symbols are not properly explained. It is suggested to have a notation section detailing the maths symbols and operations used in the paper. Eq. (3) right hand side is a bit confusing, the reviewer cannot understand that, could you explain step by step?


Minor:

-  Remove the  full stop "." inside Sec. 3.4.1 and 3.4.2 headings.

- Table of font sizes should be consistent in the whole paper. In the main paper, table 1, 2, 3 and 5 font sizes are too small, and hard to read; however, table 6 in appendix section, the font sizes are too big. The authors should make those representations consistent and nicer.

- While the maths presented in the paper seem to be accurate, its presentations, especially in the method section, needs extensive improvement. The reviewer gets lost many times, and some of the concepts are not very well introduced and explained. A good paper should explain these concepts clearly and in a more intuitive way.

- An excellent paper should use efficient figures and plots to show the experimental results. In this paper, most experimental results are presented in the form of tables, it would be better if the authors could provide some plots and visualisations to make the evaluations, comparisons and analysis more interesting to wide communities.

**Questions:**

Please refer to my main comments above.

---

> ### Author Response · Authors · 2024-11-24
> **Response to Reviewer LgjH (Major Q1-Q3)**
>
> Thank you for your thorough review and comprehensive comments.
>
> **Major:**
> > Q1: One of the major issues of this paper is that some of the concepts presented in the paper are not properly and clearly explained.
>
> A1: We have re-organized the Abstract and Section 1 (Introduction) following the reviewer's suggestions.
>
> > Q1.1: For example, in the abstract section, what is "white-box scenarios"?
>
> A1.1: We have revised the statements in the Abstract and moved the first mention of "white-box" to Section 1 (Introduction). Detailed explanations for "white-box" are now provided when it is first mentioned, as below:
> (1) Line 14: Within this research, transfer-based attack is an important tool as it mimics the real-world scenario where an attacker has no knowledge of the target model, but is under-explored in Skeleton-based HAR (S-HAR). Consequently, existing S-HAR attacks exhibit weak adversarial transferability, and the reason remains largely unknown.
>
> (2) Line 39: Existing S-HAR attacks are mainly proposed under white-box settings, where the attacker has full access to the victim model's architecture, weights, and training details.
>
> > Q1.2: Also, what does it mean by saying "its sharpness contributes to the weak transferability in S-HAR"?
>
> A1.2: We have clarified it as follows:
> Line 19: In this paper, we investigate this phenomenon via the characterization of the loss function. We find that one prominent indicator of poor transferability is the low smoothness of the loss function.
>
> > Q2: Fig.1 is not being introduced and explained clearly. For example, what is a spatial attack, spatial-temporal attack, why frozen, trainable, being lightweight and demanding? What are appended models and dual Bayesian? These concepts should be clearly explained to readers, either in texts or figure captions. Moreover, how to tell which one is better?
>
> A2: Thanks for your constructive suggestions.
> (1) Spatial attacks treat each frame independently, overlooking the temporal dependencies between sequences. In contrast, spatial-temporal attacks integrate temporal motion gradients to disrupt the spatial-temporal coherence of S-HAR models. We have added detailed explanations of these concepts to the caption of Fig.1 and further clarified their differences in Section 1 (Introduction).
>
> (2) Re-training Bayesian neural networks is computationally expensive and memory-intensive. Therefore, we freeze the pre-trained surrogate and append lightweight Bayesian components to it, turning a single surrogate into a Bayesian one without requiring re-training, thereby speeding up the training process and reducing memory footprint. This has been clarified in Section 1 (Introduction).
>
> (3) We have remade Fig.1 and added explanations in Section 1 (Introduction) to better illustrate the concept of "appended models" and the process of "dual Bayesian optimization."
>
> (4) In Fig.1 (Right), results marked with a checkmark ($\surd$) indicate superior performance compared to those marked with a cross ($\times$). This has been clarified in the captions. Additionally, we have added the transfer attack success metric to Fig.1 for clearer evaluation. The novelties of our method are also summarized to clarify its superiority in Section 1 (Introduction).
>
> > Q3:  The introduction section is not very well-motivated. For example, in Line 83, referring to Table 1; however, the concepts to understand the table are not being explained. The authors are encouraged to reorganize the section, explain the concepts properly before going deep into the technical jargon. What is "white-box"? In the contribution section, what is "low adversarial transferability"?
>
> A3: Following your suggestion, we have rewritten Section 1 (Introduction) to improve readability and provided detailed explanations of these concepts before diving into technical jargon.
>
> >Q3.1: In Line 83, referring to Table 1; however, the concepts to understand the table are not being explained.
>
> A3.1: We have revised the text to reference Table 1 in a later section to provide a detailed analysis.
>
> >Q3.2: What is "white-box"? In the contribution section, what is "low adversarial transferability"?
>
> A3.2: Detailed explanations for these terms have been added:
> (1) Line 38: Existing S-HAR attacks are mainly proposed under white-box settings, where the attacker has full access to the victim model's architecture, weights, and training details,...
>
> (2) Line 51: However, results show that their transfer success rate is highly determined by the specific choice of the S-HAR surrogate, so that its general adversarial transferability is low, also referred to as low/weak transferability.

---

> ### Author Response · Authors · 2024-11-24
> **Response to Reviewer LgjH (Major Q4-Q6)**
>
> **Major:**
> >Q4: In the related work section, the authors outline some closely related works; however, they do not highlight what are the major differences between existing works and the proposed method, what are the insights, how the proposed method has the potential to address the challenges, etc.
>
> A4: We have clarified the differences between existing works and our proposed method in Section 2 (Related Work) and highlighted the key differences and novelties in Section 1 (Introduction).
>
> >Q5: Fig.2, the font sizes are invisible to the reviewer. What do the x and y axes represent? What do "random direction vectors" mean? These concepts should be properly explained.
>
> A5: In Fig.2, the $x$ and $y$ axes represent two random direction vectors sampled from a Gaussian distribution, which are added to the model’s parameter space along these directions. These random direction vectors are used to assess the sensitivity of the model's loss function. The $z$ axis represents the loss value.  We have clarified this in the caption and increased the font size in Fig.2.
>
> >Q6: In Eq. (2), what do boldface $\mathbf{x}$ and $\tilde{\mathbf{x}}$ mean? In Line 158 and 183, it says "a clean motion" and "the adversarial sample" , what do they mean and how do they align with each other? What would be the relationship between them? These concepts and math symbols are not properly explained. It is suggested to have a notation section detailing the math symbols and operations used in the paper. Eq. (3) right-hand side is a bit confusing; the reviewer cannot understand that. Could you explain step by step?
>
> A6:
> (1) **$\mathbf{x}$** is the clean sample from the original training set $\mathcal{X}$. A white-box attack aims to find adversarial examples $\tilde{\mathbf{x}}$ within the neighborhood $\mathcal{B}_\epsilon(\mathbf{x}) = \\{\tilde{\mathbf{x}}: \\|\tilde{\mathbf{x}} - \mathbf{x}\\|_p \leq \epsilon\\}$ that misleads the target model, where $\epsilon$ is the perturbation budget and $\\|\cdot\\|_p$ is the $l_p$ norm distance. We have clarified this in Section 3 (Methodology).
>
> (2) In Eq.3, it is intractable to directly train a Bayesian component((Eq.(3) left-hand side)), so the posterior distribution $p(\theta' \mid \mathcal{D}, \theta)$ needs to be approximated through sampling (Eq.(3) right-hand side), where $p(\theta^{\prime} \mid \mathcal{D},\theta) \propto p(\mathcal{D} \mid \theta, \theta') p(\theta')$ and $p(\theta')$ is the prior of appended model weights. This has been explained in Section 3 (Methodology).
>
> We have provided inference details, mathematical derivations, and the algorithm in the Appendix. We encourage the reviewer to refer to the Appendix for additional technical details.

---

> ### Author Response · Authors · 2024-11-24
> **Response to Reviewer LgjH (Minor)**
>
> **Minor:**
> >Q1: Remove the full stop "." inside Sec.3.4.1 and 3.4.2 headings.
>
> A1: We have removed the full stop "." from the headings of Sec.3.4.1 and 3.4.2 as suggested.
>
> >Q2:Table of font sizes should be consistent in the whole paper. In the main paper, table 1, 2, 3 and 5 font sizes are too small, and hard to read; however, table 6 in appendix section, the font sizes are too big. The authors should make those representations consistent and nicer.
>
> A2:Thanks for your feedback. We have thoroughly adjusted the font sizes across all tables to ensure they are consistent and easy to read throughout both the main paper and the appendix without exceeding the page constraints.
>
> >Q3:While the maths presented in the paper seem to be accurate, its presentations, especially in the method section, needs extensive improvement. The reviewer gets lost many times, and some of the concepts are not very well introduced and explained. A good paper should explain these concepts clearly and in a more intuitive way.
>
> A3:As per your suggestion, we have provided a more detailed explanation in Sec.3.3. We hope this improves the clarity and understanding of the method. Also, we have provided the inference details, mathematical derivations, and the algorithm in the Appendix. We encourage the reviewer to refer to the Appendix for additional technical details.
>
> >Q4:An excellent paper should use efficient figures and plots to show the experimental results. In this paper, most experimental results are presented in the form of tables, it would be better if the authors could provide some plots and visualisations to make the evaluations, comparisons and analysis more interesting to wide communities.
>
> A4: We have used a radar chart to present the comparisons with ensemble attacks in Fig.3, visualized the loss surface in Fig.2, and displayed the visual results in Fig.4. In the revised version, we have also replaced the table summarizing the ablation study on motion gradient with a bar chart, as shown in Fig.5, making the evaluations and comparisons more intuitive and visually engaging.

---

> ### Author Response · Authors · 2024-11-27
> **Looking forward to further feedback**
>
> Dear Reviewer LgjH,
>
> Thank you again for your valuable comments and suggestions, which are very helpful to us. We have responded to the proposed concerns. We hope that it has helped to address the concerns you have raised in your review.
>
> We understand that this is quite a busy period, so we sincerely appreciate it if you could take some time to return further feedback on whether our responses resolve your concerns. If you have any further questions, we are more than happy to address them before the conclusion of the rebuttal phase.
>
> Best,
>
> The Authors

---

> > ### Comment · Reviewer_LgjH · 2024-11-27
> >
> > Thank you to the authors for the rebuttal. After careful consideration, I have decided to maintain my original rating.

---

### Official Review · Reviewer_BzGe · 2024-11-01

**Soundness:** 3
**Presentation:** 3
**Contribution:** 3
**Rating:** 8
**Confidence:** 4

**Summary:**

For the transfer-based attack on skeletal action recognition tasks, this paper builds the first comprehensive benchmark, comprising 7 skeletal models, 10 attack models, 3 skeletal datasets, and 2 defense methods. Additionally, a novel post-train Dual Bayesian Motion attack is presented, aiming to explore a smoother model posterior by incorporating a small MLP layer. Unlike image data, the spatial-temporal characteristics, represented by first-order velocity and second-order acceleration, are considered.

**Strengths:**

1. Extensive experiments.
2. The post-train Dual Bayesian Motion attack method is reasonable in general.

**Weaknesses:**

1. Although it is reasonable to assume that a smoother loss landscape benefits transfer attacks, exploring the reasons for poor transferability in S-HAR tasks and comparing the differences among surrogate models may reveal various factors. Therefore, directly concluding that smoothing the loss landscape is the sole solution may not be entirely logical.

2. In this method, an MLP is primarily added after the original model. However, the consideration of first-order velocity and second-order acceleration to capture spatial-temporal information, as mentioned in this paper, has already been addressed by SMART (Wang et al., 2021), which limits the novelty of this approach.

3. The Bayesian optimization approach used in this method is similar to BEAT (Wang et al., 2023), and the differences between the two approaches should be discussed further.

**Questions:**

Please see in Weaknesses.

---

> ### Author Response · Authors · 2024-11-24
> **Response to Reviewer BzGe(1/2)**
>
> Thank you for the supportive and in-depth review.
> >Q1: Although it is reasonable to assume that a smoother loss landscape benefits transfer attacks, exploring the reasons for poor transferability in S-HAR tasks and comparing the differences among surrogate models may reveal various factors. Therefore, directly concluding that smoothing the loss landscape is the sole solution may not be entirely logical.
>
> A1: We do not assume that smoothening the loss landscape is the sole solution for improving adversarial transferability. As discussed in Sec.3.1 Motivation, we first systemically show the sensitivity of adversarial transferability on the choice of surrogates. Subsequently, we compare the the loss surface smoothness of the surrogates, as shown in Figure 2, which give a clear indication of high correlations between loss smoothness and transferability. Led by this observation, we improve the adversarial transferability by properly smoothening the loss surface (not argue that smoothening loss is the sole solution). We suspect the word 'prioritize' in line 86 might have caused confusion. We have clarified this and reorganized the draft to explain the correlations between loss smoothness and transferability. Furthermore, we have added new experimental analyses of loss smoothness in Section 4.3.
>
> >Q2: In this method, an MLP is primarily added after the original model. However, the consideration of first-order velocity and second-order acceleration to capture spatial-temporal information, as mentioned in this paper, has already been addressed by SMART (Wang et al., 2021), which limits the novelty of this approach.
>
> A2: Although both TASAR and SMART consider motion dynamics, their goals and techniques differ significantly. (1) SMART employs perception loss to satisfy the visual naturalness in white-box settings, while TASAR leverages Bayesian motion gradient to disrupt the spatial-temporal co-herence, thereby improving attack transferability in black-box scenarios. The latter approach is more feasible in challenging, real-world scenarios, in which white-box information is not attainable. (2) The technical frameworks of the two methods also differ fundamentally. SMART maximizes attack loss only using position gradient while preserving perception constraints by formulating the attack as an optimization problem:
> $$
> \begin{equation}
>     \mathop{\min}\limits_{\delta} L(\mathbf{x}+ \delta, y) + \lambda d_p (\mathbf{x}, \mathbf{x}+ \delta),
> \end{equation}
> $$
> where $L$ is the classification loss and $d_p$ denotes the perception loss, which penalizes changes in dynamics by constraining the first-order velocity and second-order acceleration using the $L_2$ norm distance. In contrast, TASAR maximizes attack loss using Bayesian motion gradient instead of naive position gradient under a small $l_\infty$-ball constraint. Specifically, TASAR employs time-varying autoregressive models (TV-AR) to explicitly model temporal motion and compute its corresponding high-order gradients. Also, SMART does not incorporate motion dynamics into Bayesian framework. To further assess the impact of these technical differences, we conducted an ablation study by substituting perception loss for the motion gradient in TASAR (denoted as TASAR+PL) and comparing it with TASAR using the motion gradient (TASAR+MG). The results demonstrate that integrating perception loss into TASAR significantly reduces attack performance compared to using the motion gradient. We have further clarified the difference in paper.
>
> **Table 1: The attack success rate (%) of untargeted transfer-based attacks on NTU60. “Ave” was calculated as the average transfer success rate over all target models except for the surrogate.**
>
> | Surrogate    | Method     | STGCN | FRHEAD | MSG3D | CTRGCN | Ave   |
> |--------------|------------|-------|--------|-------|--------|-------|
> | **ST-GCN**   | TASAR+PL   | **100.00** | 44.53  | 60.93  | 17.17  | 40.88 |
> | **ST-GCN**   | TASAR+MG   | 99.29 | **49.41** | **64.60** | **20.33** | **44.78** |
> | **CTRGCN**   | TASAR+PL   | 27.34 | 57.81  | 57.03  | 94.54  | 47.39 |
> | **CTRGCN**   | TASAR+MG   | **33.76** | **58.32** | **66.74** | **97.06** | **52.94** |

---

> ### Author Response · Authors · 2024-11-24
> **Response to Reviewer BzGe(2/2)**
>
> >Q3: The Bayesian optimization approach used in this method is similar to BEAT (Wang et al., 2023), and the differences between the two approaches should be discussed further.
>
> A3:  Although both BEAT and TASAR use the post-train Bayesian strategies, their respective goals, formulation and optimization are completely different.
>
> (1) BEAT aims to defend against adversarial attacks, whereas TASAR aims to generative adversarial examples with high transferability to fool black-box skeleton classifiers. Their goals are fundamentally opposite.
>
> (2) Their formulations differ significantly. BEAT jointly models clean data, adversarial distributions and classifier to enhance adversarial robustness. TASAR introduces Gaussian noise into the model's parameters during the sampling process to smoothen the surrogate posterior. This smoothing process ensures that the loss surface becomes smoother, which can enhance adversarial transferability.
>
> (3) Their Bayesian optimizations also diverge. Unlike BEAT using a naive post-train Bayesian optimization(only updating $\theta'$), TASAR proposes a new Post-train Dual Bayesian Optimization to smooth the appended network weights (simultaneously updating both $\theta^{\prime}$ and $\Delta \theta^{\prime}$). Different from BEAT only updating $\theta^{\prime}$, TASAR entangled optimization of $\theta^{\prime}$ and $\Delta \theta^{\prime}$ presents a more complex gradient updating challenge.
>
> As shown in Table 4 in Sec.4, compared to the vanilla Post-train Bayesian strategy, Post-train Dual Bayesian strategy significantly improve the attack performance. Following reviewer's suggestions, we have further clarified these distinctions in the paper.

---

> > ### Comment · Reviewer_BzGe · 2024-11-26
> >
> > The author has addressed my concerns and agreed to increase the score.

---

> ### Author Response · Authors · 2024-11-26
>
> We sincerely thank you for your positive feedback on our paper! If you have any questions or suggestions, please let us know!

---

### Official Review · Reviewer_NGYv · 2024-11-02

**Soundness:** 3
**Presentation:** 4
**Contribution:** 2
**Rating:** 6
**Confidence:** 4

**Summary:**

This paper studies transfer-based attacks for skeleton-based action recognition. A last-layer Bayesian approches is applied to smooth the loss surface to improve adversarial transferability. The authors setup a benchmark for transfer-based skeleton-based action recognition attacks. The proposed method is evaluted using different backbone approaches on NTU-60, NTU-120, and HDM05. The results demonstrate the effectiveness of the proposed approaches.

**Strengths:**

1) The paper is well-written with clear motivations of each step
2) The proposed approch is easy to apply on existing methods since only a few layers are needed at the end of the networ.
3) Thorough experiments and ablation studies

**Weaknesses:**

1) As the pretrained models are frozen, modeling p(\theta’|D, \theta) is not as good as modeling p(\theta, \theta’|D). If trainable, it would be better to justify the performance difference on a small subset of data.
2) Besides the visualization of loss surfaces, quantitative evaluation of the surface smoothess can be added.

**Questions:**

1) In Table 4, why K=3 yields better results than K=5? I thought more samples lead to better approximation

**Details Of Ethics Concerns:**

No concerns

---

> ### Author Response · Authors · 2024-11-24
> **Response to Reviewer NGYv(1/2)**
>
> Thank you fou your valuable feedback and generous suupport.
> > Q1: As the pretrained models are frozen, modeling $p(\theta'|D, \theta)$ is not as good as modeling $p(\theta, \theta'|D)$. If trainable, it would be better to justify the performance difference on a small subset of data.
>
> A1: We conducted additional experiments to compare the performance of modeling $p\left(\theta^{\prime} \mid D, \theta\right)$ , $p(\theta \mid D)$  and  $p\left(\theta, \theta^{\prime} \mid D\right)$. The default setting of TASAR corresponds to $p\left(\theta^{\prime} \mid D, \theta\right)$, where $\theta^{\prime}$ is trainable while $\theta$ remains fixed. We use the proposed post-train dual Bayesian optimization to update $\theta^{\prime}$. $p(\theta \mid D)$ represents a standard Bayesian neural network without adding appended models. For this case, we train the Bayesian surrogates with a cyclical variant of Stochastic Gradient Markov Chain Monte Carlo[1] to sample 3 models from the posterior distribution of neural network weights. For $p\left(\theta, \theta^{\prime} \mid D\right)$, 9 appended models are added behind BNNs and both $\theta$ and $\theta^{\prime}$ are trainable. Due to the high optimization complexity(updating both $\theta$ and $\theta^{\prime}$), we use vanilla post-train Bayeian optimization instead of improved post-train dual Bayesian optimization optimization for updating $\theta^{\prime}$.
>
> The results are presented below. Compared to modeling $p(\theta \mid D)$ and $p(\theta, \theta'|D)$, TASAR achieves superior performance in transfer attacks on three out of five black-box models and demonstrates the best white-box attack performance on ST-GCN. This advantage arises from the use of improved post-training Dual Bayesian optimization, which enables smoothed posterior sampling and improves adversarial transferability. Moreover, unlike modeling  $p(\theta \mid D)$ and $p(\theta, \theta'|D)$, where averaging gradients from multiple surrogates diminish white-box attack effectiveness, our post-training strategy preserves the pre-trained model intact, not reducing the original white-box attack strength. Further, TASAR significantly reducing computational overhead and accelerating training process, as shown in Table 2. So we use $p\left(\theta^{\prime} \mid D, \theta\right)$ as our default setting. We have added the experimental analysis in the appendix.
>
>
> **Table 1：The attack success rate(\%) of untargeted attacks on NTU60. ’SFormer’ represents SkateFormer.**
> | Surrogate     | Method                                     | ST-GCN  | 2s-AGCN | MS-G3D | CTR-GCN | FR-HEAD | SFormer |
> |---------------|--------------------------------------------|---------|---------|--------|---------|---------|---------|
> |    | $p(\theta'\|D,\theta)$(TASAR) | **99.29** | **42.55** | 64.60  | **20.33** | **49.41** | 17.22   |
> |    **ST-GCN**            | $p(\theta\|D)$                           | 90.63   | 37.11   | 69.53  | 17.58   | 43.36   | **25.39** |
> |               | $p(\theta, \theta'\|D)$                      | 93.75   | 41.80   | **73.04** | 18.75   | 46.87   | 22.65   |
>
>
> **Table 2： Model Size and Training Time measurement on NTU60. 'MS' means The Model Size(M) and 'TT' means The Training Time(hours).**
>
>
> | Surrogate    | Method                 | MS    | TT   |
> |--------------|------------------------|-------|------|
> |              | $p(\theta'\|D, \theta)$  | 3.54  | 0.65 |
> | **ST-GCN**   | $p(\theta\|D)$           | 9.30  | 5.7  |
> |              | $p(\theta, \theta'\|D)$  | 9.36  | 6.4  |
>
> References:
> [1] Zhang et al, Cyclical Stochastic Gradient MCMC for Bayesian Deep Learning, ICLR 2020

---

> ### Author Response · Authors · 2024-11-24
> **Response to Reviewer NGYv(2/2)**
>
> > Q2:  Besides the visualization of loss surfaces, quantitative evaluation of the surface smoothness can be added.
>
> A2: Following your suggestions, we quantitatively measure the changes in loss as $\mathbf{x}$ is perturbed along a random direction with varying magnitudes. Specifically, we first sample $\mathbf{d}$ from a Gaussian distribution and normalize it onto the $\ell_2$ unit norm ball as $\mathbf{d} \leftarrow \frac{\mathbf{d}}{\|\mathbf{d}\|_{F}}$. Then, we calculate the loss change (smoothness) $f(a)$ for different magnitudes $a$:
>
> $$
> \begin{equation}
> f(a) = \left | L(\mathbf{x} + a \cdot \mathbf{d}, y, \theta) - L(\mathbf{x}, y, \theta) \right | .
> \end{equation}
> $$
> We calculate $f(a)$ 20 times with different randomly sampled $\mathbf{d}$, and take the average results. For fair comparison, we use the same $\mathbf{d}$ each sampling in both 'NT' and 'TASAR'. The experimental results reveal that TASAR achieves a significantly smoother loss landscape compared to Normally Training (NT) across all magnitudes of perturbation. For large perturbations (\(|a| = 1.0\)), TASAR reduces the loss change by approximately threefold compared to NT. Additionally, TASAR maintains a more uniform smoothness across different magnitudes, while NT exhibits sharper variations, with larger loss changes. This indicates that TASAR effectively smoothens the loss landscape, contributing to improved transferability. We have added the experimental analysis in the appendix.
>
> **Table 3：Loss changes ($f(a)$) measurement for normally trained surrogate and TASAR on HDM05. "NT" means "Normally Training".**
>
> | Surrogate    | Method  | -1.0  | -0.8  | -0.6  | -0.4  | 0.4   | 0.6   | 0.8   | 1.0   |
> |--------------|---------|-------|-------|-------|-------|-------|-------|-------|-------|
> |       **ST-GCN**        | NT      | 7.46  | 6.05  | 4.34  | 2.08  | **2.06** | 4.33  | 5.91  | 7.24  |
> | **ST-GCN**   | TASAR   | **2.66** | **1.70** | **1.03** | **1.77** | 2.27  | **1.40** | **1.65** | **2.31** |
>
> >Q3: In Table 4, why does $K=3$ yield better results than $K=5$? I thought more samples lead to better approximation.
>
> A3: From a Bayesian perspective, there is an infinite number of ways to draw the classification boundaries, each differing in adversarial transferability. Theoretically, as noted by the reviewer, sampling a larger number of models could enhance adversarial transferability. However, due to computational overhead, the number of sampling is constrained and small ($K\leq5$). Consequently, each appended model significantly influences the final expected performance. For $K=5$, we suspect that the two additional models fail to provide meaningful diversity and instead introduce redundancy, such as overlapping decision boundaries, which diminishes overall performance. Further increasing the number of sampling may improve the adversarial transferability, but it will cause extra computation overhead. When $K=3$, TASAR has already outperformed the baseline methods, so we choose $K=3$ and do not try more samplings.

---

> > ### Comment · Reviewer_NGYv · 2024-11-26
> > **Rebuttal feedback**
> >
> > Thanks authors for the rebuttal, I am satisfied with the answers.

---

> > > ### Author Response · Authors · 2024-11-27
> > >
> > > We sincerely thank you for your positive feedback on our paper! If you have any questions or suggestions, please let us know!

---

### Author Response · Authors · 2024-11-24
**Summary of Changes**

Dear ACs and Reviewers,

We sincerely appreciate the reviewers for their valuable and insightful suggestions, which are helpful for us to improve our work. We have carefully considered each issue mentioned in these comments and revised our manuscript based on them.  The revised version of the manuscript has been submitted, with the corresponding changes highlighted in blue. Below, we address each comment from the reviewers individually.

---

### Meta-Review · Area_Chair_7zgo · 2024-12-23

**Metareview:**

The paper presents a well-written, easy-to-apply approach for transfer-based attacks for skeleton-based action recognition. The new approach improves adversarial transferability and addresses limitations in S-HAR attacks, demonstrated by thorough experiments and ablation studies. On the other hand, reviewers have major concerned about the unclear explanations of concepts like "white-box scenarios" and technical jargon. The paper is advised to better motivate its sections and clarify its unique contributions compared to existing work. Some other minors include figures and equations that lack clarity and are not sufficiently introduced, causing confusion about their significance.

**Additional Comments On Reviewer Discussion:**

Two reviewers are satisfied with the responses, while one reviewer maintains the original rating. There is a minor divergence in the overall ratings, but the revision and changes in the manual were thorough and already addressed most of the reviewer's concerns. For that reason, an "Accept" decision is recommended for this paper.

---

### Decision · Program_Chairs · 2025-01-22

Accept (Poster)